# Mechatronics: A Study on Its Scientific Constitution and Association with Innovative Products

**Ana Carolina Cintra Faria and Sanderson César Macêdo Barbalho** *

Post-Graduation Program of Mechatronic Systems, Department of Industrial Engineering, Technology College, University of Brasilia, Campus Darcy Ribeiro, Brasilia CEP 70910-900, DF, Brazil; anacarolina.cintrafaria@gmail.com
* Correspondence: sandersoncesar@unb.br

**Abstract:** Mechatronics as a science is a synergic combination of mechanical engineering, electronic control, and software design in product development and manufacturing processes. To understand how the field of knowledge that incorporates mechatronics in innovative products, given that it is not in itself a basic engineering discipline but an integration of fields of knowledge, has advanced, it was developed a bibliometric and qualitative study through systematic review with an analytical framework for the establishment of variables to subsidize the construction of the selected theoretical body. The results and conclusions of the sampled publications show that mechatronics performs one of the principal roles in innovation due to the multidisciplinary integration that the scope of innovation in product engineering is propitiating. The study classified five global scenarios: practical approaches aimed at product development, research that studies curricula and education in engineering, studies involving components of a mechatronic system, use of artificial intelligence, and methodologies for designing mechatronic systems. In addition to underscoring that the use of the term innovation associated with mechatronics in a large proportion of the publications extrapolates the operational level, characterizing an attribution to the term that is always associated with the applications, ramifications, and perspectives that the respective product, design, robot, or system could offer to the market or future research. Similarly, it was found that the results of many publications associate the term innovation with a return on investments or operational costs and emphasize the advantages of using the technology for commercial ends.

**Keywords:** innovation; mechatronics; mechatronic system; bibliometric analysis; systematic review; analysis framework

## 1. Introduction

In recent years the literature on Big Data, Data Science, and their applications in the analysis of scientific publications has grown fast, boosting bibliometric studies to obtain a visualization made possible by the deep mining of data, revealing a panorama of information that is essential for decision making in a given area [1].

The term mechatronics originated in Japan, and the term itself indicates the fusion and integration of mechanics, electronics, and software for the miniaturization of systems and mechanisms and, finally, the integration of control systems for the adjustment of mechanical and electronic parameters according to the external application conditions. Mechatronics as a Science can be defined as a synergic combination of mechanical engineering, electronic control, and software design to develop products and manufacturing processes. It is related to systems, devices, and product design projects that seek to achieve an ideal equilibrium between the basic mechanical structure and its general control [2,3].

When researching the conceptual construction of mechatronics, there are few studies on the substantial and essential factors for analyzing and mapping the field of mechatronics science, mechatronics engineering, and mechatronic systems, mainly associated with innovation.

Since it began in the late 1970s, mechatronics has configured itself as a necessary approach and integration to enable innovation and boost manufacturing processes in Japan. Machines and tools with numerical commands occupy outstanding positions in the technological fusion that predominates in the industrial automation and electric motor sectors [4]. Ref. [5] notes that the Japanese companies' increasing investment in Research and Technology has repercussed in the form of a series of technological fusions that have become the standard way of generating innovations.

While formerly it was sufficient to understand electricity, mechanics, and automation, it can increasingly be seen that the concept of innovation is inherent and indissociable when the subject is mechatronics. The area has been configured according to the advances achieved in industry, automation, and artificial intelligence. It is familiarized with new technology, virtualization, big data, the internet, and computing in the cloud, for example. Innovations are described, analyzed, and tested to optimize processes and make products more and more competitive. A closer investigation of that scenario reveals that there is a growing number of publications related to the research applied in the publicizing of results of the components of a mechatronics system, of products or services that are considered innovative; on the other hand, it can be seen that little attention has been devoted to understanding the mechatronics concept as a research field or its strong connection with the concept of innovation [6,7].

To add to the scenario presented above, this article sets out to conduct a basic study to amplify and strengthen the base of knowledge that lies at the heart of mechatronics, exploring the tendencies that are in the vanguard of its development and seeking answers to the set of problems under study.

To that end, Information Science offers the necessary mechanisms to reveal, through the association of methods, the reference parameters for analysis that will map the knowledge structure in that field of research. The methods used to analyze the scientific collaboration recommended by Information Science will make it possible to associate the origins of the mechatronics concept with innovation and to do so in the framework constructed for its analysis.

The literature considers interdisciplinary quantitative measurements a dimensional approach in science studies [8]. In this article, we adopt bibliometrics, and qualitative and quantitative methods, constituting what is known as combined research. Using those methods enables better mapping and contextualization of where research on the theme is being conducted, the predominant research type, and the form of its dissemination, in addition to identifying what is being produced.

It is necessary to highlight that although considered an asset and mentioned in works such as [9] and important definitions of innovation, mechatronics needs to be studied more to understand its constitution when considered an attribute of innovation. In works with some aspect of an approach similar to this research, it is always the asset that characterizes products [10], an inseparable attribute in course curricula [11,12].

Based on the assumption and discussions in the literature that mechatronics doesn't have a single dominant definition, as it is an interdisciplinary science resulting from the integration of mechanics, electronics, and software, and considering the birth history of the area intrinsic to the Japanese practice of integrating electronics to mechanical machine tools to make improvements in production processes, it is possible to observe that this practice can now be considered innovation [13–15]. Therefore, the research problem that guides this article is to investigate how the elements that constitute the field of knowledge that subsidizes the construction of mechatronics are treated from the perspective of their association with innovation.

Being the innovation a very relevant attribute present in mechatronic constitution as a science, researching your association enable us to understand and get to know how a science develops from attributes such as innovation and its lateral areas.

And the central objective of this article is to analyze the constitution of the field of mechatronics. Considering the multidisciplinary basis of mechatronics, our effort was cen-

tered on the relationship between mechatronics and innovation. The authors' background pointed out that most mechatronics applications, in synergy with the first publications on the term "mechatronics", are attempts to innovate by mixing electronics with mechanics and software. So, it means that in the beginning, innovation was an attribute of mechatronics, or putting differently, mechatronics is an attempt to innovate. Therefore, our initial point was a framework for mechatronics and innovation from classical literature. Utilizing quantitative bibliometric analysis, it was possible to identify scientific collaboration networks, and by analyzing the sample publications with the initial framework, we mapped the main elements of mechatronics and the innovation perspectives that it reinforces in the scientific literature for the two largest available databases, Web of Science and Scopus.

The quantities related to this research objective are (i) the range of publications in the main databases Web Of Science and Scopus by "mechatronic*" in the conventional indexing fields: title, keywords, and summary, resulting in a total of 33 thousand documents, (ii) due to the assumption that innovation is an attribute, an asset in mechatronics when we associate "innovation*" the overall result is 854 (eight hundred and thirty-four) publications, which makes the sample proposal feasible for bibliometrics, systematic review, and qualitative methods intended.

The article has been organized into sections. First, the main theoretical elements of the research are presented. Then, the method applied to the research, the classification protocol, and the framework used to analyze and classify publications are presented. In the third stage, the results are presented. We identify the phases described in the "Integrated methodological flow of scientific analysis and mapping" to present the bibliometric analysis results and the qualitative/classificatory analysis built from the analysis framework built to conduct the research. Next, the results of the bibliometrics and the qualitative analysis based on the framework are presented, along with the conclusions, limitations, and contributions for future studies. The set of references used for our bibliometric analysis is presented in the supplementary file to this article.

## 2. Mechatronics as a Science

There are various approaches to and discussions of the definition of mechatronics as a science, ranging from the very conception and emergence of the term to its current applications. Some articles in the literature report that in Japan in the late 1970s, Mechatronics appeared as an interdisciplinary area combining Mechanical and Electronic Engineering with Computer Science [16]. Ref. [17] consider that the successful combination of mechanics and electronics embraces the origin of the term 'mechatronics' and digital processing in consumer products. Some studies mentioned how Professor Takashi Kenjo used the term informally for years. Still, the first commercial use of the term mechatronics applied to the industry is attributed to Tetsuro Mori in 1969 [18].

Ref. [19] defines the area of mechatronics as a combination of Mechanical Engineering, Control engineering, Microelectronics, and Computer Science in a concurrent engineering approach. That means they must have a simultaneous vision of the technical problems and the possible solutions in the different disciplines involved, in contrast to the traditional approaches that usually address the problems separately, discipline by discipline.

In the 1950s, the industrial and process control markets began to use the mechatronics concept, whereby the mechatronics system was characterized as a process in which sensors capture information from the physical world that is digitally processed, resulting in control actions. The control system acts on the physical system through actuators. The overall result is a feedback system that may involve systems with varied levels of complexity [17].

It was in the 1980s that countries like Australia, Japan and South Korea, and various European countries created the first undergraduate and graduate courses offering the multidisciplinary teaching of Mechatronics [20].

With the advent of computing as a decision-making tool, those definitions needed to change, considering the increasingly apparent amplitude of the systems of the mechatronics context. Examples include traditional mechanical systems such as machine tools,

vehicle manufacturing and control automation, process automation systems, and thermal, environmental, and building vibration control systems [21,22].

While the original concept of mechatronics focused on integrating Mechanical Engineering, electronics, and the use of software, later, that emphasis was found to need a more holistic vision that might include system design and development. The change of emphasis effectively placed mechatronics amid a network of engineering functions and issues ranging from aesthetics to marketing. When reviewing that network, it is important to acknowledge and understand that, generally speaking, the mechanisms of engineering designs do not concern the use of technology alone but also depend on people and the interactions among individuals [21,23].

Given the progress made in all those areas that depend on the integration of those sciences that compose mechatronics, the definitions of the term could no longer solely consider the integration of specific technologies; it was necessary to adopt an approach orientated toward systems for the design, development, and implementation of complex systems.

In his study 'Mechatronics—more questions than answers', Ref. [21] questioned whether mechatronics continues to be an area that should or must continue to be separated and distinct from other engineering approaches and engineering projects or whether the proposals, studies, and progress achieved are sufficient for it to become incorporated into conventional engineering, a question to discuss.

Ref. [24] conducted a study based on a review of the literature on the evolution of mechatronics definitions on the analysis of students in areas related to the application of mechatronics in industrial projects, and they discuss what has been developed over the last 50 years and what could be expected in the future for mechatronics as a science.

They made a case study identifying how engineers participating in the respective project were responsible for integrating knowledge based on a 'combination of areas' from the perspective of developing a software product while the engineers in other areas focused their work on "their respective areas of responsibility". The study's conclusion points to future discussions of convergence between the software engineers' activities and those of the mechatronic engineers. In that case, there would not be a replacement of mechatronics by software engineering. Still, the latter could become an essential tool for supporting mechanical and electrical engineering designs offering greater global control over the decision-making for mechatronic systems in real-time. Another important point they discussed is why mechatronic engineers have worked as software engineers in developing embedded systems [24].

The increasing progress in integrating intelligent systems conceived based on artificial intelligence stood out in reviewing the recent mechatronics literature. Furthermore, just as mechatronics developed in the integration of various disciplines, so there is a need for education in mechatronics to have, as one of its fundamental bases, the integration of various areas, especially "since the solution of real problems invariably requires integrating different subjects and disciplines, both technical and non-technical" [25].

Ref. [26] discuss the influence of the Industrial Internet of Things (IIoT) on mechatronics development. Based on the various definitions of the term mechatronics, the authors underscore the re-signification of the definition, which includes information systems as one of its main parts. Thus, mechatronics must be considered a "bridge between a world orientated by information and the physical world, based on its integration with the IIoT. Among the tendencies in mechatronics studies, the authors underscore the topics of intelligent devices, precision, new materials development, and miniaturization (nanotechnology), which require the overcoming of innumerable challenges in the qualification of new engineers and the development of mechatronics design processes [26].

Throughout its history and development, mechatronics has acted as an important, sometimes essential differential factor in integrating with other areas and elaborating new products. Even though it is a constituent part of mechatronics as a science, that integration always deserves an outstanding section in studies based on integrating mechatronics with other disciplines and technologies [27,28].

### 3. Innovation

One of the literature's classic and best-known definitions of innovation is that of [29]. The author makes a distinction between the concepts of invention and innovation. The invention is "[...] an idea, an outline or model for a new or improved artifact, product, process or system". On the other hand, he perceives innovation as an evolutive system insofar as it is necessary to alter the production system, incorporate new functions and ways of organizing the work. The results of those changes are new products and improvements in the existing products or processes. That is to say; innovating makes it possible to open new market niches. Schumpeter establishes those definitions in the economic context, which inevitably presupposes the characterization of innovation as being associated with commercial exploitation, given that companies are the foundation of economic development [29].

Historically, 'innovation' has been mainly associated with positive aggregated values to qualify knowledge, objects, and processes. The word innovation has been well accepted by society but with different conceptions, reflected in the more than fifty similar and different definitions offered by various dictionaries and papers [30,31].

In the previous section dealing with the various conceptions of mechatronics, it is already possible to discern how the concept of innovation is at the heart of the discussions of the term, especially in the aspect of being based on the integration of the different areas responsible for promoting improvement in development, costs and the applications of products and services, for example. Some authors address the question of innovation based on Mechanical Engineering and its tradition in developing innovations within the automobile industry [32–34]. According to [32], "software is the driving force of innovation in mechanical engineering", and modern mechanical systems are developed based on the integration of different branches of engineering and accordingly designated as mechatronic systems. Thus, integrating various areas, the conceptual base of mechatronics is also considered to be the innovation responsible for elaborating complex systems such as software.

Those different forms of innovation extend to a varying extent to different market teams, departments, and branches. Although there is some overlapping of the various definitions of innovation, the number and diversity of the definitions give rise to discussion and divide researchers whose works emphasize how difficult it is to define the term [31,33,34].

At the heart of those varying concepts of innovation, there are the following premises: that it is a novelty introduced into the market that aggregates value to that already existing; the creation or modification of a product/system and its introduction in the market; the process of translating an idea or an invention into goods or a service that creates value to the end consumer. Science often appropriates those definitions to describe the transfer of research and development results, the classic sequence of the transfer of knowledge and technology to the market. However, the use of the term, whether it is to produce Science for use in an industrial context or even as it is used by society and companies, carries with it a background that is inseparable from development, research, and something new which can be considered the 'links' among the isolated compartments of innovation types [31,33–36].

According to the OSLO Manual, which established the guidelines for gathering and interpreting innovation data, the term can be characterized as the implementation of a new (goods or service) product or a significantly improved one, or of a new marketing method, or a new organizational method in business practices, the organization of the workplace, or external relations [33].

Based on the presupposition that innovation is a technical term often used erroneously, one of the world's leading authorities on innovation [34,35] explains that the use of the term innovation has become generalized and non-descriptive and it is, therefore, necessary to specify clearly and objectively what is being denominated as innovation.

One type of innovation, known as 'disruptive innovation', has become one of the main competencies for modern organizations operating in competitive globalized markets. Companies are increasingly studying and implementing the radical changes associated

with the concepts or technologies involved in process and product development that seek to obtain product quality or ergonomics to conquer new Markets and maintain those they have already conquered [36–38].

In the last few years, there has been a notable change from systems based on the interconnection of physical components in which the transmitted data have been used to facilitate control to systems in which the information is at the heart of the system and is served by intelligent objects for integration of precision mechanical engineering, electronic and intelligent computerized control has led to the emergence of a technological area with widespread application and one that is indispensable for the development of innovative products and for the innovation of manufacturing processes. It is an area of great expansion in present-day and future industries and requires specialized personnel. Automation is also one of the first steps for the industry to incorporate technology to optimize processes, thereby making its products more competitive [39–43].

Observing the context of the historical evolution of mechatronics and the studies that indicate how engineering professionals are very flexible and quite capable of handling different solutions in an agile and efficacious manner, thereby making mechatronics a strategic career and, added to that, the various elements that corroborate the inseparability of the pathways of the concept of innovation and the progress of mechatronics then, albeit, not explicitly, the mechatronics-innovation association is a source of research only secondarily noted in a good part of the publications mentioned so far in this article when the review of curricula, the adaptation of methodological processes and mechatronic science are set as the analytical perspectives [42–46].

## 4. Methods and Data

This section describes the stages of classifying the scientific publications, data analysis, and the methods used. The integrated flows of the methodology used in the research are shown in detail in Figure 1, and the stages and types of analysis are described in the sections that follow the topic.

The set of methods employed is intended to achieve the proposed objective and afford a vision of the results of the analyses, the analyses' academic impacts, the publications collaboration network, and the focus of each of the methods that the analyses employed to support responses to the mapping of how the scientific field embracing innovation in mechatronics is developing.

Each of the stages displayed in Figure 1 is described in detail below.

### 4.1. STAGE 1—Articles Classification Protocol

Stage one, identified in Figure 1, corresponds to a description of the terms used in the search strategy and the selected specifications in each database.

Scientific publications were selected in the ISI Web of Knowledge, a database offered by the Institute for Scientific Information (ISI), and in the SCOPUS, the Elsevier Publishing House reference database. Without delimiting the year of publication before the year of the database search (2021) and considering titles, Abstracts Keywords, and indexing as being the Taxonomic, Systematic and Descriptive terms, according to Table 1.

After the definition of the terms, the advanced search strategy was to map any types of publication that contained the words or phrases mechatronic, mechatronics engineering, or mechatronics systems associated with the word innovation and, after tests, the definition of the best strategy resulted in a total of 890 publications.

Figure 2 illustrates the steps taken to select the articles. The defined advanced strategy was executed in the Web of Science and Scopus databases, resulting in 890 publications with no restriction regarding the year of publication or documental typology. After eliminating redundancies, 760 publications remained. After applying the legibility criterion, the complete text's availability resulted in 646 publications for analysis.

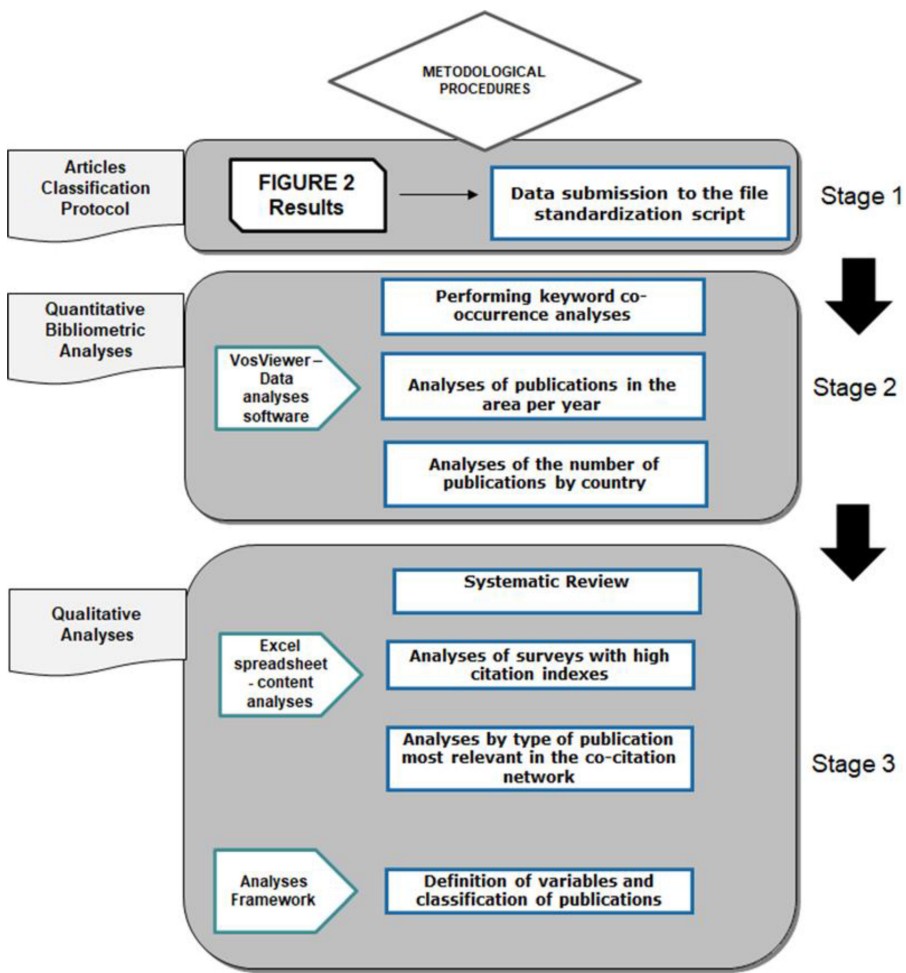

**Figure 1.** Integrated scientific mapping and analyses methodology flows (source: adapted from [47–49]).

**Table 1.** Terms and Boolean operators used in the advanced search (source: own elaboration).

| Bases and Specifications | |
|---|---|
| Data Base | **Scopus e Web Of Science** |
| Publication Period | **Until 2021** |
| Boolean operators: | **"AND" "OR" " * " "( )"** |
| Type of publications: **All** | |
| Keyword combinations | |
| Textual content:**TITLE-ABS-KEY-AUTH** | |
| Terms: **"mechatronic*" "mechatronics engineering" "mechatronics systems " "innov*"** | |
| Final combination: **WEB OF SCIENCE** **TS = ((innov*) AND (mechatronic OR "mechatronics engineering" OR "mechatronics systems"))** **SCOPUS** **TITLE-ABS-KEY (innov* AND mechatronic OR "mechatronics engineering" OR "mechatronics systems")** | |

The 646 publications were tabulated in an Excel spreadsheet with their descriptive data and the qualitative attributes identified in stages 2 and 3. To combine the publications in a single file and use them as input for processing in the VOS Viewer software (version 1.6.15), a php script was developed to standardize the inputs to a single format. Details of the script are set out in the results section.

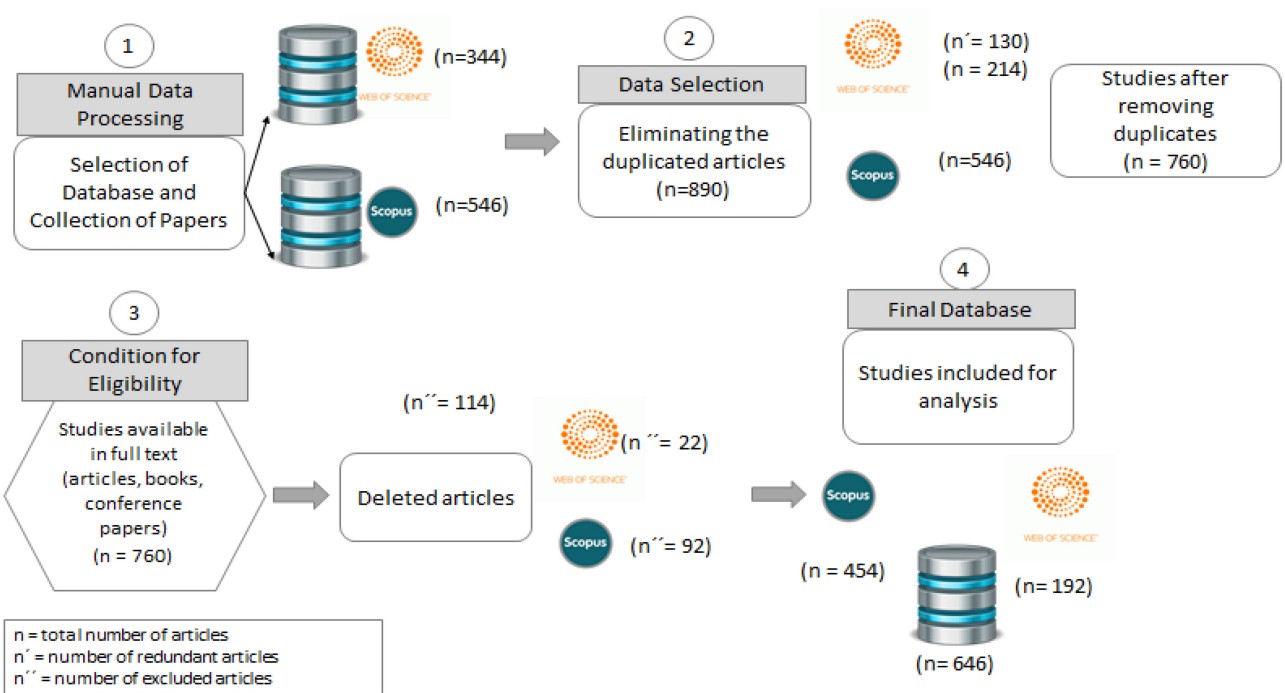

**Figure 2.** Article selection protocol (source: own elaboration).

*4.2. STAGE 2—Bibliometric Analyses*

Quantitative and qualitative bibliometric analyses were used to analyze the sample (constituting stage 2). Bibliometric studies are a quantitative scientific research method, and their first definitions go back to the beginning of the 21st century. A sampling of the definitions and discussions on the attributions of variables and the enhanced outreach of the terminology makes it possible to characterize bibliometric studies concerning the aspects of: (i) measurement of author productivity based on a size-frequency distribution model applied to various researchers in a pre-defined set of documents, (ii) measurement of the frequency with which keywords appear to obtain the relationship between the terms of a given discipline or subject, and lastly (iii) measurement of the productivity of the journals, establishing the nucleus and areas of dispersion for a given subject in the same set of journals [50–57].

To be able to conduct the analyses, it was necessary to compile the data extracted from both the Web of Science and Scopus. The two databases' extraction processes made it impossible to unite all the sample publications in a single file to be submitted to the VOS Viewer in this study. The two use different taxonomies to identify the classes and attributes of the categories, which made it necessary to standardize them in a single format.

That being so, based on the interpretation of the data and the recurrence of patterns in the files, the researchers developed a script in PHP (an open-source script language) that transforms files into a single pattern/standard. The developed web application was of fundamental importance for standardizing the data, eliminating redundancies, and submitting all the sample publications to a single file.

The quantitative analyses used bibliometrics to analyze the co-occurrence of keywords, the number of articles per year, and the number of publications per country and to quantify other parameters, such as citations, to subsidize the qualitative analyses.

Among the various areas embraced by the study and application of Information Science, one is dedicated to studying the relations in virtual communication networks and, outstandingly, to those established among scientific researchers. In the so-called collaboration networks, the production of interconnected individuals is considered, reflected in the final research product: scientific communication. In other words, the evolution of

scientific development, irrespective of the area, can be estimated by analyzing the scientific production of a group of researchers or a country regarding a given subject [58,59].

This study used the VOS Viewer software to process and visualize the bibliometric data. The software enables creating and visualizing biometric maps based on a large volume of data. And is used to construct maps of authors or journals based on citation data, keyword maps based on co-occurrence data, and to create maps of title words or abstract words on journals, articles, co-authorship, and bibliographic coupling. The program offers a visualizer that makes it possible to examine bibliometric maps in detail, emphasizing different aspects of the same map [9,60].

Even in stage 2, we already carried out bibliometric and classificatory qualitative analyses to categorize the publications and identify qualitative aspects. The parameter used to determine the qualitative analysis sample was the Pareto principle. In the distribution by the Law of Pareto, also known as the Laws of Power, the hypothesis that 80% of the effects stem from 20% of the causes was stated [61,62].

The qualitative parameters of the data analysis were applied in 20% of the 246 most cited articles and submitted to the systematic review's qualitative criteria. Characterized by making available a summary of the evidence associated with a specific intervention strategy employing explicit, systematized search methods, critical appreciation, and synthesis of the selected information, the systematic review is an effective method for selecting the content of the kind of sample involved in this study. This method is useful because it integrates the information of a set of studies in a given area, undertaken separately, demonstrating the results of the analyzed publications [63]. Systematic literature reviews classify the scientific contributions of the area that is the object of study and, therefore, require a much greater data-gathering and analysis effort than traditional reviews. This kind of content reduces the possibility of bias being introduced by subjective interpretation that can occur with traditional reviews [64–66].

### 4.3. STAGE 3—Qualitative and Framework Analyses

The literature in which the object of study and analysis of this article is inserted presents a wide and complex range of variables that can be analyzed to delineate an investigative panorama of the concepts and presuppositions that we set ourselves: it is not possible to concretely and explicitly identify in it that various innovative products that had given rise to important brands and innovative companies can be classified as mechatronic prospects. However, as that is not presented clearly, it configures a gap that needs to be explored, which gave rise to the present study.

Table 2 presents the analysis framework constructed to support the classifications and qualitative attributions developed in Stage 3 of the research.

The constructed framework constitutes stage 3 and is divided into variables: product development, technological development, components in a mechatronics system, and mechatronic areas. Each one was stratified in levels to provide an understanding of the physical field of the research studies that discuss the concept of innovation in mechatronics based on consolidated concepts, models, and research on the literature.

The analysis variables range around the product concept, which is a rough definition, and is an element that connects companies and their clients. Product development is a process of transforming data from markets and technology into new knowledge to generate products and services in commercial production [67,68]. It can be defined as a mechatronic product when it is possible to observe, in its development, the integration of mechanical, electronic, and software technologies and that its basic functions are supplied by integrating the technologies that compose it. Mechatronic products are also considered complex ones insofar as alterations in scope during project development are frequently observed [6,68]. Ref. [4] describes the characteristics of mechatronic systems and products as being functional, mechanical, and electronic interaction among information technologies; spatial integration of subsystems in a physical unit; intelligence related to the control functions of the mechatronics system; flexibility and easiness with which the mechatronic products can be modified to meet new

requirements and address new situations; multifunctionality attributed to the software functions defined by the microprocessor; invisible functions performed by microelectronics and difficult for clients to perceive or understand; and technological dependence in strict connection with the available industrial technologies.

**Table 2.** Classification and analysis framework (source: own elaboration).

| | Analysis Variable | Analysis Level | Authors |
|---|---|---|---|
| Product Development | By innovation project | Product Technology Process Technology | Wheelwright and Clark (1992), Clausing (1994), Clark and Fujimoto (1991), Ulrich and Eppinger (1995) Wheelwright e Clark (1992) |
| | By type of technology used (degree of project innovation) | Incremental Platform Radical | Clark and Guy (1998), Henderson e Clark (1990); Burgelman, Maidique e Wheelright (1998) e Christensen (2019) |
| Technological Development | Advanced research and development | | Wheelwright e Clark (1992), |
| Components in a Mechatronic System | Product structure and technologies involved | Sensors and Instrumentation Processing and Control Software Actuators and Drivers Engineering Project Communication system | Barbalho (2006); Bradley (2000); Bernardi et al., (2002) |
| Areas of mechatronics | By main subjects | Mechanical Engineering Electrical engineering Materials Engineering Chemical engineering Control System Embedded Systems Software Engineering Math Neuroengineering Agronomy | Bradley (2015); Azar (2019); Isermann (2008) |

Although projects that can be considered innovative are far from forming a homogeneous class, a series of typologies are used to classify them, the most outstanding being that of [66], which distinguishes them according to the degree of innovation conceived in the project. Those authors classify the attribute of innovation according to the degree of novelty present in the product and its production process. They suggest four main types: incremental, platform, radical, and advanced Research and Development innovation.

The term 'innovation project' when discussing process technology refers to the process the company defines when it develops a new product. All the operations use some process technology, which helps the production meet a clear market need and, on other occasions, becomes available. Also, a given operation may adopt it to exploit its potential. The tools, machines, and equipment that help the production transform materials, information, and consumers to aggregate value and achieve the objectives of strategic production strategies make up the so-called process technologies. They are also considered instruments responsible for the mediation between inputs and outputs and constitute a set of knowledge accumulated in generating products and services. On the other hand, the use of technologies embedded in a product is considered product technology. Its value aggregation configures the company in the market context, that is, the use of new technologies to produce equipment, materials, software, and the inputs used in developing the product [66,69–71].

According to [66], aggregate project plans can be classified according to the degree of change they propose regarding product or manufacturing. They divide the projects into five groups: alliance projects, derivative projects, platform projects, breakthrough projects, and advanced development projects. The alliance or partnership projects are based on associating with another organization to develop a new product. The so-called derivative projects normally require fewer resources and are sustainable as they are roughly

variations of the same platform. The platform or new generation projects involve more dimensions insofar as they need to be better than the preceding platforms and derivatives and require greater skills, creativity, time, and resources of the people involved. Such projects normally produce a new family of products. The products that call for radical changes, known as breakthrough projects, establish new key processes and products that present significant changes.

Thus, one of the classification categories is associated with the type of technology employed, and it may be radical, incremental, or of the platform type. From the perspective of the technology used, the degree of innovation is widely discussed in the literature on the influence of management on the product development process [66,69,72]. Within that analysis framework to attribute the degree of innovation, the proposal of [66] presented earlier in conceptualizing the so-called aggregate project plan, which considers the following categories of innovation: radical, incremental, platform, or new generation.

The research and advanced technology variable refers to developing technological knowledge for future applications, usually without direct commercial objectives. Together with alliance projects, this variable, in terms f technological development, seeks to develop know-how associated with the technology used in the previously described degrees of innovation and is concentrated on the fast, objective launching of products for commercial ends.

Another level of the analysis framework concerns the components in a mechatronics system, the product structure, and the technologies involved. The framework structure has been based on the work of [68], who developed a classification based on the technologies cited by [73,74]. Ref. [68] defines a mechatronics system's main components as sensors and instrumentation, processing/control software, actuators and drives, basic engineering design, and communication system. The attributed classification and the presentation of the systematization of the technologies/components involved in a mechatronics system are distributed in the analyzed article's sample.

In his classification, Refs. [21,22] did not include neuro-engineering, mechanical, chemistry, materials engineering, or agronomy. Still, they are present in the work of [37,38] and are considered strategic areas that characterize expressive publication niches. So, they have been incorporated and classified by subjects. In general, these technological fields are related to the concept of basic engineering design [68].

The last analysis variable used in Stage 3 consists of the main mechatronic subjects that [21] attributed importance to, given their aspect of thematic scientific knowledge areas used in the mechatronics courses and previously presented by the same author in 1997 and updated in 2015. It is necessary to consider that classification because of the conceptualization difficulty that mechatronics has always experienced in the academic milieu and among professionals engaged in industry, given the divergence among the approaches and emphases on its identity in the scientific community. Because of that difficulty, Bradley states that qualified engineers must have the specific integration skills provided by mechatronics education and qualification. In other words, it is necessary to balance detailed knowledge and the ability to act as an integrator of a wide variety of environments [21]. In that context which Bradley [21,22] classified, the areas and sub-areas of the respective mechatronics curricular program were divided into control systems, embedded systems, analogical and digital software engineering, and added to them, thematic attributions such as mechanical engineering, neuro-engineering, chemistry, and materials engineering.

## 5. Results

The subsections corresponding to each presented stage discuss the results of the analysis of each one of them.

### 5.1. STAGE 1

To begin the bibliometric analyses, it was first necessary to create a PHP script to unite the files with the published articles (eliminating redundancies) from the Scopus and Web

of Science databases. It was not possible to submit files simultaneously to the VOS Viewer software used to analyze the scientific collaboration aspect as the characteristics of the files extracted from the two databases are different. So, it was necessary to find a solution to convert the files to a single standard. The script registers and searches for blocks for conversion, transforming the textual information of the file extracted from Scopus into the abbreviations used by the Web of Science.

*5.2. STAGE 2*

The quantitative bibliometric analyses were carried out to gather more objective evidence to describe the scientific progress and how mechatronics is developing in the contexts of products and innovating perspectives. As explained in the methodology section of this paper, the bibliometric analyses were used to provide data that could demonstrate and support the analysis results, review types of publications, year of publication, and origin of what has been discussed and achieved in terms of the scientific production of the area; and help us to understand the thematic areas and research predominance using the keywords. In total, 646 publications were subjected to bibliometric analysis.

5.2.1. Co-Occurrence of Keywords

The files extracted from the Scopus database, after being processed according to the classification protocol of this research, were submitted to the script created to standardize the format of the data in the VOS Viewer and were then submitted concomitantly with the Web of Science file for processing and the generation of the co-occurrence map displayed in Figure 3.

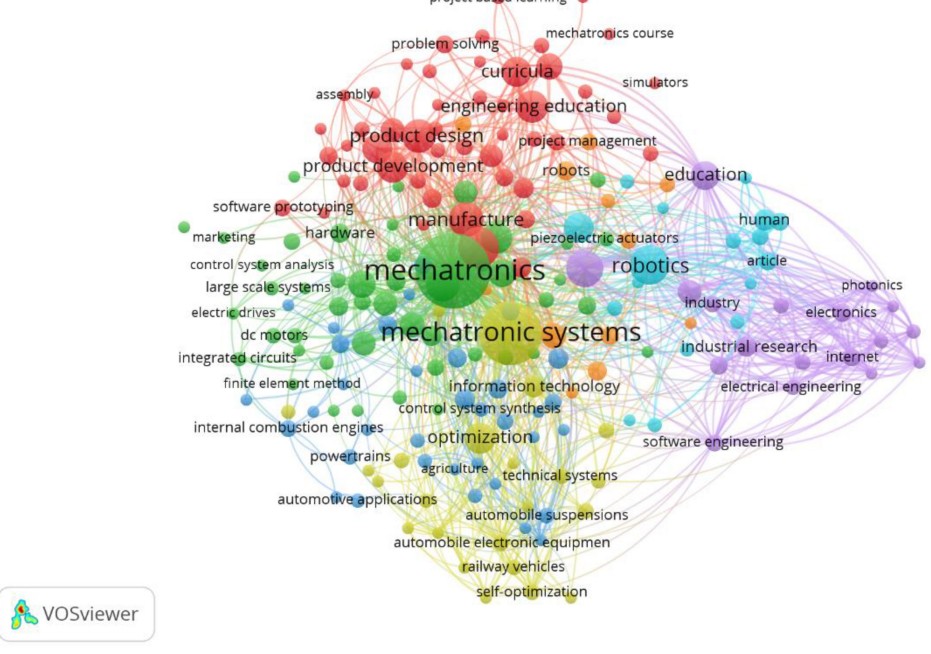

**Figure 3.** Map of the co-occurrence of keywords processed in the VOS Viewer software (source: own elaboration).

The nodes in the co-occurrence map represent the weight of an item (words), and the circle of an item shows the size of the node in the network; the greater the weight of an item, the larger its circle and the larger the corresponding label. Connections and the nearness or farness of the circles correspond to the occurrence of the words in the same publication. Observing the most frequent ones among the sets of words (bigger nodes) is possible. The clusters of the same color on the map belong to the same knowledge area, so it is also possible to visualize the most recurrent themes and their connections and easily extract the keywords that have the greatest influence on the networks being analyzed.

Thus, the number of keyword occurrences and, consequently, their weight and degree represented in the size of the network node enables us to categorize the publications clusters of Figure 3 based on their respective colors in the map as follows:

- Red Cluster: the publications are concentrated in the 'manufacture' area, especially computer-assisted, using software to control machines and equipment tools related to the manufacturing process, such as Computer Assisted Design CAD) and Computer Assisted Manufacture (CAM). The nodes in the red area with the greatest weight embrace the works of [75–78] and represent the areas of 'product design', 'mechatronics products', and 'product development'. Those nodes are connected to less expressive ones such as 'software prototypes', 'mechatronics product development', 'software project management', and 'simulators'. Also, in this first cluster, a group of works studies 'engineering education', 'the 'curricula' of 'researchers' and 'undergraduate courses', 'engineering disciplines', 'knowledge acquisition', and 'problem-solving'.

- Green cluster; publications such as those of [79–81] are examples of the green cluster and are concentrated in the areas of 'machine design', 'actuators', 'control system analysis', 'sensors', 'mathematical models', integrated circuits, large-scale systems', 'intelligent systems', 'robotics systems' and accident prevention'.

- Blue Cluster: the nodes in the blue cluster are smaller and more scattered, which indicates the lesser incidence of the respective terms in the sample, and they consist mainly of publications in the area of the automobile industry ('electro-mechanical devices', 'automobile manufacture', 'suspension', 'complex systems', 'mechanical actuators', 'machine components', 'cinematics') and optimization applications, especially for the automobile sector. Examples of publications are those of [82–84].

- Yellow Cluster: in the yellow cluster, the works of [85–87] are identified by the terms 'mechatronics systems', associated with 'optimization', 'electronic industry', learning systems', 'information technology', 'self-optimization', 'energy management', 'suspension components', 'electrical energy', 'electronics' and 'energy storage'.

- Purple cluster: In the purple cluster, the heaviest node is 'education' associated with 'industrial research', 'electrical engineering', 'software engineering', 'internet', and other less expressive subtopics, but also in the industrial field as can be seen in the works of [88–90]. One of the most expressive nodes of this cluster, whose visualization was suppressed in the best visualization extracted from VOS Viewer, represented in Figure 3, is an educational program.

- Light Blue Cluster: In this cluster, the 'robotics' node is central and the weightiest, strongly associated with the term 'humans'. Another frequently appearing in this cluster is 'mechatronics devices', and although smaller nodes do not appear in the figure, they can be observed in the network using node zoom mechanisms in the VOS Viewer. Important exponents of this cluster are the works of [91–93].

- Orange Cluster: the orange cluster is the smallest and least expressive. The terms in it are 'robots', 'actuators', 'automation', 'control', 'information management', 'laboratories', 'mechanics', 'mobile robots', 'monitoring', and 'product development process'. Unlike the term 'robotics' in the blue cluster, the term 'robots' in the orange cluster expresses those works that address the question of robot development based on case studies such as those of [94–96].

Terms that were not included in the search strategy are considerably well-represented. Examples are 'manufacture', 'design', and 'engineering education', representing areas of expressive importance within the scope of the study sample.

5.2.2. Publications in the Area by Year of Publication

The second quantitative analysis was of the distribution of the articles by year of publication, as illustrated in Figure 4. No temporal limitations were imposed in either of the databases (Scopus and Web of Science), and the earliest registration of an article in this area dates to 1984. In the sixteen years from 1984 to 2000, there needs to be more expression of articles on innovation mixed with mechatronics, with 1999 marking the year with the

most, just seven articles. However, in the first year of the 21st century, articles rose to 13. There were some historical occurrences of publications at the end of the 20th century but in other knowledge areas. They had some factors in common, addressing high technology use, mechatronics, robotics, and high-performance computers in milestone events in the historical evolution of technology in the context of mechatronics. For example, in 1997, the first successful cloning of a mammal was achieved; a computer scored a victory over the world chess champion—the computer Deep Blue defeated Boris Kasparov in a new match of six games, of which two the computer won, 3 were a draw, and in one it was defeated. It was the first computer to defeat a world champion in a match run according to official time rules. There was also the event of the first mobile robot on Mars equipped with optical cameras and an X-ray spectrometer to make a chemical analysis of the Martian soil [97].

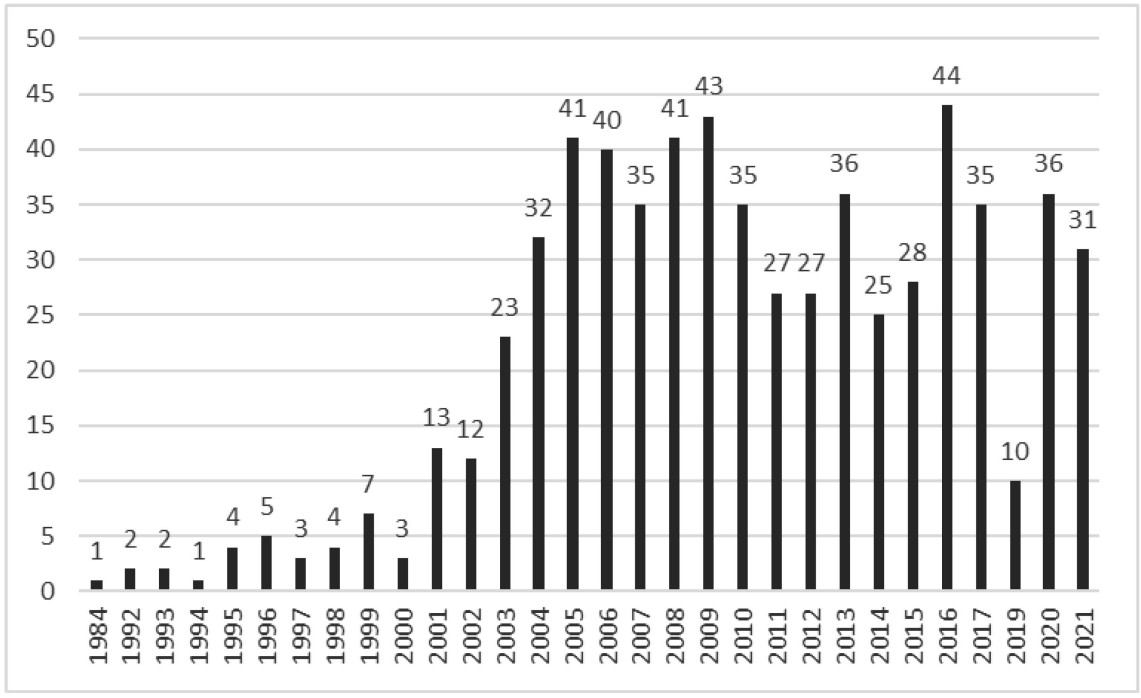

**Figure 4.** Distribution of articles per year (source: own elaboration).

From 2003 on, the number of publications began to increase. While technology was changing in support, size, and intensity, the North American, European, and Japanese industries aggressively employed high-quality, low-cost engineering services regardless of the international frontiers. Thus, technological and market changes and changes in teaching and the application of mechatronics needed to be reviewed. The publications analyzed in the global scenario when filtered for the time interval around the turn of the 20th to the 21st century, register considerable discussion of the undergraduate and graduate courses in the various branches of engineering [98–105].

The last two years in our sample (2020 and 2021) were marked by publications in which the terms 'Internet of Things' (IoT) and 'Industry 4.0' appeared frequently associated with the terms 'research' and the context of applying the analyzed variables. In 2019 there was a decrease in the number of publications, but a general analysis, the average of publications in the context of the theme, has been increasing since 2001.

### 5.2.3. Number of Publications per Country

In Figure 5, the publications have been distributed according to the country of publication and the respective numbers. The countries that published the most were Germany (177 publications), Italy (102 publications), the United States (58 publications), and China (44 publications).

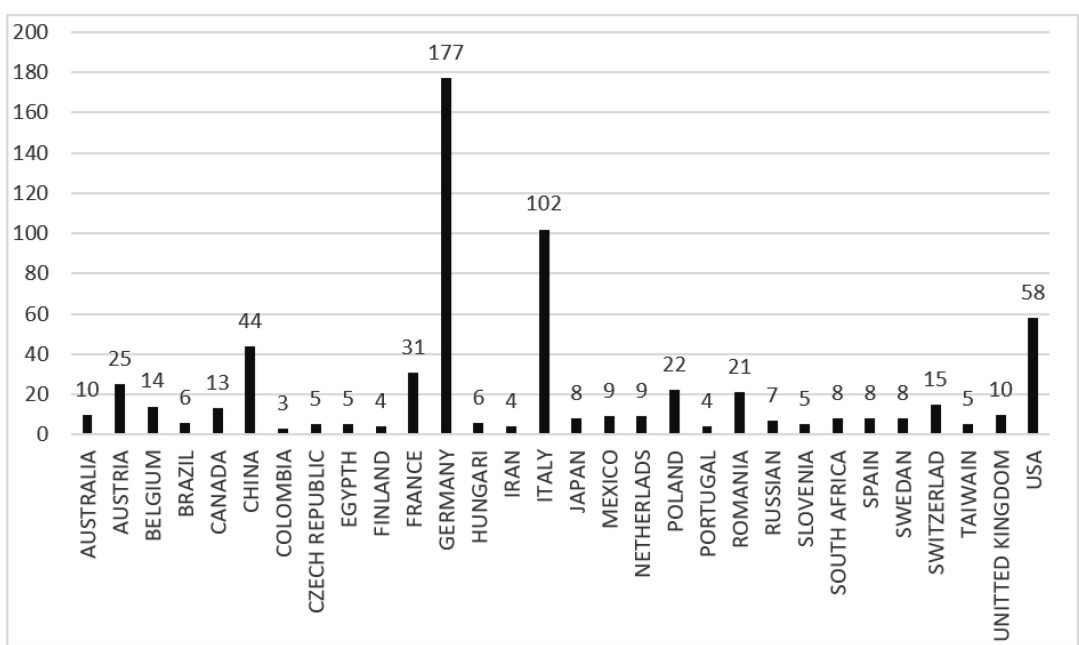

**Figure 5.** Distribution of the articles by countries (source: own elaboration).

By default, databases such as Web of Science and Scopus provide the authors' country, university, and department of affiliation just below the title. And for distribution by country, the Web of Science bases field was considered, which associates researchers with their respective countries and institutes, universities, and affiliations. It should be noted that this article analyzed the country of publication and not the collaboration between the publication's authors.

These countries' publications address numerous themes and aspects of innovation in mechatronics, and some of the more outstanding approaches can be summarized here. China concentrates its main work on robotics with applications in health and automation. The United Kingdom has the most diversified publications involving mechatronic systems, complex systems, mechatronic products, electronic systems for vehicles, engineering education, and undergraduate course curricula. The Italian publications make significant contributions to the automobile industry and those of the United States on automobile manufacture, internal combustion engines and equipment, progress, and the need for change in the curricula for the mechatronics courses.

5.2.4. Research Publications with High Citation Numbers

From here on, we used the Pareto Principle to select the publications analyzed qualitatively 246 publications, corresponding to 80% of the most cited ones, were used.

It can be seen that among the 13 studies with the highest impact on the citations network displayed in Table 3, only two are actually in the citations network, the studies of [106,107].

There was an expressive number of self-citations whereby the authors mention and dialogue with their own earlier research which was identified in qualitative terms as being the continuation or intersection of the research undertaken by those authors. For example, it can be seen in the research studies of [106,108,109] that in 2012 his study concerned an innovative mechatronic system to control the hand's tendons. In 2013, the continuation of that research proposed a design solution aimed at simplifying, reducing the costs, and improving the system's robustness for controlling the hand's tendons. In 2014 the continuity of that research showed the integration of the finger of the hand, the sensors, and the actuation, with special attention paid to the sensor subsystem's requirements and the hand's mechanical structure. In other words, the self-citations are associated with prior research studies whose constitution is indispensable for the development of the current

study. That is the case with all the works in the network in which self-citation is observable. It can be inferred that the high number of self-citations is related to the great variety of areas that characterize mechatronics, whereby an author has a beginning in his basic area. His references are more those of his basic area than specifically those of mechanics.

**Table 3.** Research publications with a high number of citations (source: own elaboration).

| Author | Title | Source | Number of Citations | Abstract |
|---|---|---|---|---|
| Diftler et al. (2011) | Robonaut 2-the first humanoid robot in space | Proceedings—IEEE International Conference on Robotics and Automation | 246 | NASA and General Motors developed the second-generation Robonaut, Robonaut 2 or R2, a state-of-the-art, dexterous anthropomorphic robotic torso with significant technical improvements over its predecessor, making it a far more valuable tool for astronauts. The R2's integrated mechatronic design results in a more compact and robust distributed control system. |
| Massa et al. (2002) | Design and development of an underactuated prosthetic hand | Proceedings—IEEE International Conference on Robotics and Automation | 233 | This article focuses on an innovative approach to developing prosthetic hands based on underactuated mechanisms. Furthermore, it describes the development and a preliminary analysis of a prototype of an underactuated prosthetic hand. |
| Isermann (2008) | Mechatronic systems-Innovative products with embedded control | Control Engineering Practice | 231 | This contribution summarizes ongoing developments for mechatronic systems, shows design approaches and examples of mechatronic products, and considers various embedded control functions and the system's integrity. One field of ongoing developments, automotive mechatronics, where especially large influences can be seen, is described in more detail by discussing mechatronic suspensions, mechatronic brakes, active steering, and roll stabilization systems. |
| Bracewell and Sharpe (1996) | Functional descriptions used in computer support for qualitative scheme generation—"Schemebuilder". | Artificial Intelligence for Engineering Design, Analysis, and Manufacturing: AIEDAM | 132 | With mechatronic product development as the main theme, this paper describes a closely integrated methodology incorporating a bond graph approach to continuous-time energetic systems and high-level Petri nets for the rigorous description of discrete-time information systems. |
| Grebenstein et al. (2010) | Antagonistically driven finger design for the anthropomorphic DLR hand arm system | IEEE-RAS International Conference on Humanoid Robots, Humanoids 2010 | 85 | The paper presents a finger design that combines a reduced diversity of parts with the need to build five kinematically different fingers. The fingers are protected against overload by allowing subluxation of the joints. The tendon routing allows for an antagonistic actuation and is optimized to minimize friction and wear. |
| Horowitz et al. (2007) | Dual-stage servo systems and vibration compensation in computer hard disk drives | Control Engineering Practice | 80 | This paper discusses two mechatronic innovations in magnetic hard disk drive servo systems, which may have to be deployed soon to sustain these devices' continuing 60% annual increase in storage density. |

**Table 3.** *Cont.*

| Author | Title | Source | Number of Citations | Abstract |
|---|---|---|---|---|
| Gouaillier (2009) | Mechatronic design of NAO humanoid | Proceedings—IEEE International Conference on Robotics and Automation | 77 | This article presents the mechatronic design of the autonomous humanoid robot, the NAO, built by the French company Aldebaran-Robotics. It distinguishes itself from existing humanoids thanks to its pelvis kinematics design, its proprietary actuation system based on brush DC motors, and its electronic, computer, and distributed software architectures. |
| Merzouki et al. (2012) | Intelligent mechatronic systems: modeling, control, and diagnosis. | Springer Book | 69 | Presents the recent developments in the design of intelligent mechatronic systems. Offers practical advice on designing practical, functional, and safe intelligent systems. |
| Palli et al. (2014) | The DEXMART hand: Mechatronic design and experimental evaluation of synergy-based control for human-like grasping | The International Journal of Robotics Research | 68 | This paper summarizes recent activities to develop an innovative anthropomorphic robotic hand called the DEXMART Hand. This research aims to face the problems that affect current robotic hands by introducing suitable design solutions to achieve simplification and cost reduction while possibly enhancing robustness and performance. |
| Goth, Putzo and Franke (2011) | Aerosol Jet printing on rapid prototyping materials for fine-pitch electronic applications | Proceedings—Electronic Components and Technology Conference | 57 | Printing technologies allow producing fine-pitch electronic applications to miniaturize mechatronic systems further. Aerosol Jet $^®$ is an innovative non-contact and maskless printing process for fine structures below 50 µm with the possibility to process a wide variety of inks, including nanoparticle inks on different substrate materials. This research aims to qualify the Aerosol Jet technique to manufacture prototypes for MID (Molded Interconnect Devices). |
| Constantino et al. (2011) | Design and Test of an HV-CMOS Intelligent Power Switch with Integrated Protections and Self-Diagnostic for Harsh Automotive Applications | IEEE Transactions on Industrial Electronics | 53 | The design and characterization of high-voltage (HV)-CMOS technology of an innovative, intelligent power switch (IPS) for harsh automotive applications is proposed in this paper. The electrical simulations, experimental characterization, and testing at component and onboard system levels prove that the proposed design allows a compact, smart power switch realization facing the harshest automotive conditions. |

**Table 3.** *Cont.*

| Author | Title | Source | Number of Citations | Abstract |
|---|---|---|---|---|
| Allotta et al. (2015) | Preliminary design and fast prototyping of an Autonomous Underwater Vehicle propulsion system | Journal of Engineering for the Maritime Environment | 52 | As a partner of THESAURUS, the Mechatronics and Dynamic Modelling Laboratory of the Department of Industrial Engineering, University of Florence, the project has developed an innovative low-cost, multi-role autonomous underwater vehicle, Tifone. This article deals with the adopted methodologies for autonomous underwater vehicle design: in particular, the main focus of this study is related to its propulsion system. |
| Huck et al. (2005) | Tutorial about the implementation of a vehicular high-speed communication system | International Symposium on Power Line Communications and its Applications | 51 | The growing network of mechatronic components within a vehicle and their interaction in creating new systems request special and higher challenges to future communication systems. Besides the software implementation of these innovative systems, investigating new physical layers is of essential interest and special interest in this article. |

Another observable aspect is the great heterogeneity of the content for the researched theme, ranging from the earliest interlocutors of the article who dialogue with the area and appear in the citations network through the significant contribution of citations in numerical terms. In other words, it was found that the construction of mechatronics associated with innovation as a science is widely discussed in practical terms and with different approaches. Still, they relate very little to the works on its constitution and consolidation as a science. Another important point is that of the 246 articles that were analyzed; only 19 appear in the citations network, which means that there are very few authors who dialogue in the area on the subject and many authors, 227 altogether, who dialogue in other networks that were not identified as they lay outside the scope of the present research.

5.2.5. Analysis of the Citation Network of Publications

According to [110], analyzing a network of citations allows us to identify its most influencing studies. Then a network of document citations was built to show the works cited at least thrice once the network did not present changes with the parameters between 3 and 5 occurrences. The statistical data obtained through a citation network allow us to define some intellectual structures of the researched area [111]. The node's size in the network differentiates the concentration of citations in which the center of the network shows the most recurrent studies and identifies their relationships [9].

If we take as a basis the most cited studies of the sample, the Pareto principle was applied for qualitative analysis, where 246 publications corresponded to 80% of the number of most cited publications in the sample; the network of citations between them has a greater connection, as shown in Figure 4.

It is possible to infer that when observing the sample 646 publications, the authors do not associate mechatronics and innovation in their key terms in the publications, and the theme studied here is adjacent. However, when we reduce the scope for the 246 publications analyzed, the expressiveness of the citation network changes and becomes denser. And by analyzing when a study simultaneously cites two other distinct works (co-citation) in Figure 6, it is possible to identify that the network is much denser, besides being related to the analysis of the most cited publications in Table 2.

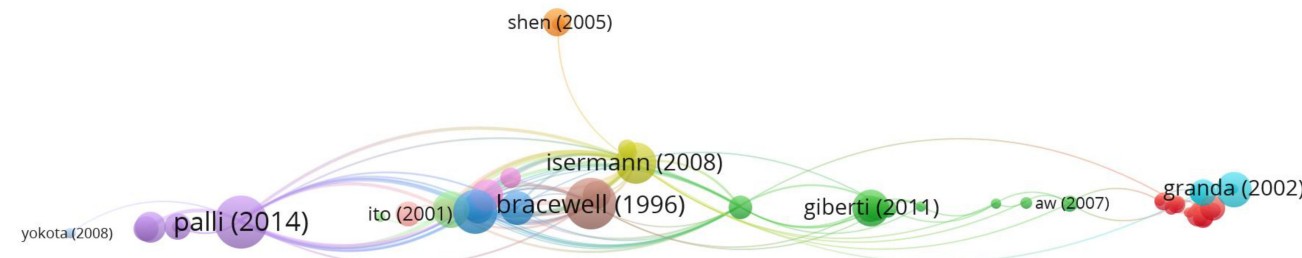

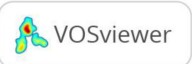

**Figure 6.** Citation network of the most cited documents in the sample (source: own elaboration).

To perform the final analysis of the citation network of the most cited publications, the minimum number of 5 citations per author was established since the citation network map is more clearly configured with parameter 5, generating a total analysis of 18 authors of the 246 total studies analyzed. It is possible to observe that the works of [76,106,112], which are among the most cited, appear in the network of citations with the expressiveness of the nodes in the network.

Despite being a small network, it is verified that these previously mentioned publications are presented as classic authors of the area, which are present in the publications regardless of the time frame because they bring definitions and important basic concepts in mechatronics.

Another network analysis that showed strong expressiveness is co-citation, which occurs when a work simultaneously cites two other distinct works (co-occurrence of citation). These works are highly cited and usually represent works by renowned authors in the field [113].

Based on the published documents, the co-citation seeks to analyze, identify and describe the structure and connectivity of an area of scientific knowledge [114]. Co-citation analysis is used to identify how often two items, be they documents or authors, from the previous literature, are cited together by some item in the more recent literature [115]. Two publications (or two authors) are considered co-cited when a third cite them. Thus, the greater the number of documents in which two authors (or two publications) are co-cited, the stronger the co-citation relationship between them [9]. This analysis allows us to identify and describe the structure and connectivity of an area of scientific knowledge [114] via published documents.

The same author also defines that, when measuring the strength of co-citation between two documents, the degree of association between pairs of documents is evidenced, according to the understanding of the community of citing authors, as they are recognized by their peers [115].

In this network of co-citation of Figure 7, five clusters are observed. In its main nodes, two of them are the publications of Bradley (red and purple), Iserman and Hehenberg (yellow), Cristen (green), Vogen—Heuser (blue); by the nodes and amount of total publications of the sample, the density of collaboration is intermediate. But the collaboration density is high when we analyze the total sample with all types of publications.

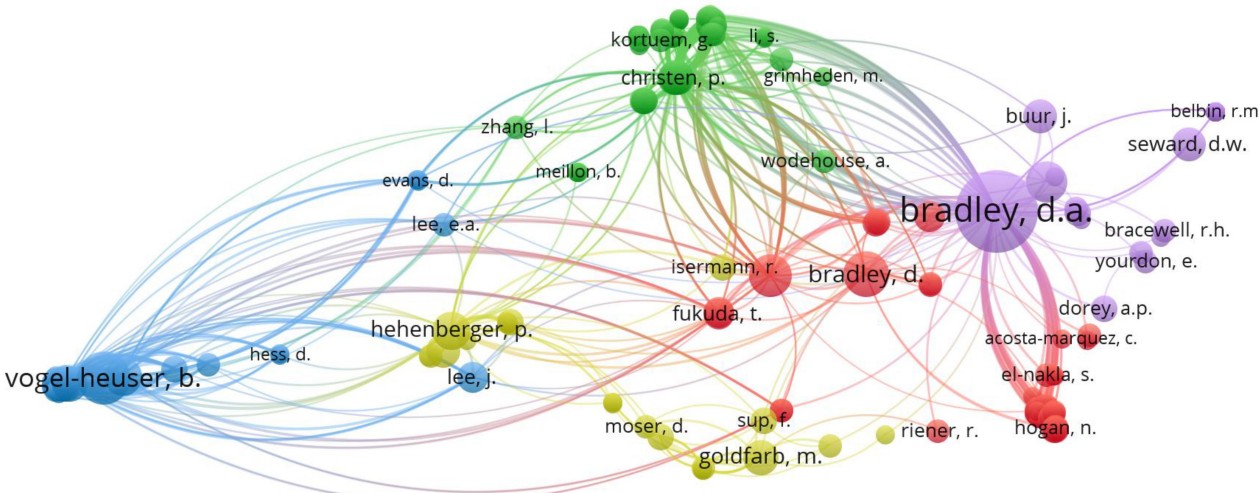

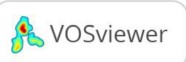

**Figure 7.** Co-citation network of the most cited documents in the sample (source: own elaboration).

In the co-citation map, it is evident that the premise of co-citation proposed by [115–117] that the proximity and interlocution of two documents are not determined by the authors of the documents but rather defined by the scientific community that appropriates their content, establishing connections during the process of generating new knowledge. Bradley is one of the authors who base the premises and pillars from which the assumptions of this thesis emerged; Isermann, Buur, and Palli also bring definitions and aspects that consolidate not only definitions and premises of the review of the state of the art, as the construction of the areas of mechatronics that supported the analyses carried out in this thesis. These same authors make up the co-citation network of the thesis sample.

The proximity between the clusters is the sum of the distances from the node, considering all the others, and serves as an efficiency index because it is interpreted as the time until the arrival of the knowledge that flows through the network. Figure 5 shows a greater distance between the green, red, yellow, and blue clusters. It is also possible to identify that the two clusters with Bradley (purple and red) as the central node are linked to the works that are also recurrent among the most cited in the sample [76,112].

If we look at the 246 most cited works, 12 are books, 96 are articles published in journals, and 138 are publications in congresses, conferences, and seminars.

The intensity of co-citation between two references is determined by the number of publications in which both articles are cited. At the time of publication, the two articles may appear not to be linked. Their links may appear (and grow over time) when these articles are jointly cited in the scientific literature. Thus, the strength of co-citation is determined by the reaction of researchers to published articles [118]; that is, it evidences the knowledge structure of a given area according to the understanding of the citing community. And although the parameter of time frame is a determinant for scientific publications, it is observed that these publications of the network of co-citations, even extrapolating the periods commonly used as a time frame (5, 10 years), are present in the scope of the works developed during so many years.

To ensure that we are dealing with the scientific production of the same author, it is necessary to consider all possible entries in primary and secondary sources. David Allan Bradley, a single author, has two distinct ways of entering his name in the analyzed sample: Bradley D. and Bradley D.A. As extracted from Scopus and Web of Science, these author entries are valid and used to designate you in publications, so there are two clusters Bradley

D. and Bradley D.A. associated with a single entry for author David Allan Bradley. Whose research underlies the variables of the qualitative analysis framework.

Although it is possible to change the author's name to a single format in the VOS Viewer input file, we chose to keep both formats because: (i) it shows that the same author can be counted differently, which consists of at least point of attention in bibliometrics to understand when this happens; (ii) if we change the file submitted to VOS Viewer in the author field due to the reduction in the number of characters, it reduces the cluster configuration, by deleting the way the author was cited in other works.

Therefore, it was decided to maintain the original configuration, where the two Bradley clusters are arranged to designate their publications.

*5.3. STAGE 3*

In stage 3, the 246 publications were read and classified according to the analysis framework presented in Table 2.

5.3.1. Product Development by Innovation Project

Of the 246 publications, 91 are publications that develop, apply or describe some process technology that helps production meet a clear market need, according to Figure 8. Publications classified as product development and innovation project, in which product technologies highlight standards and specifications related to composition, properties, and quality requirements. Those in which process technologies are developed or adopted, procedures for combining inputs and basic means for producing goods and services, including process manuals, maintenance manuals, and quality control, are presented.

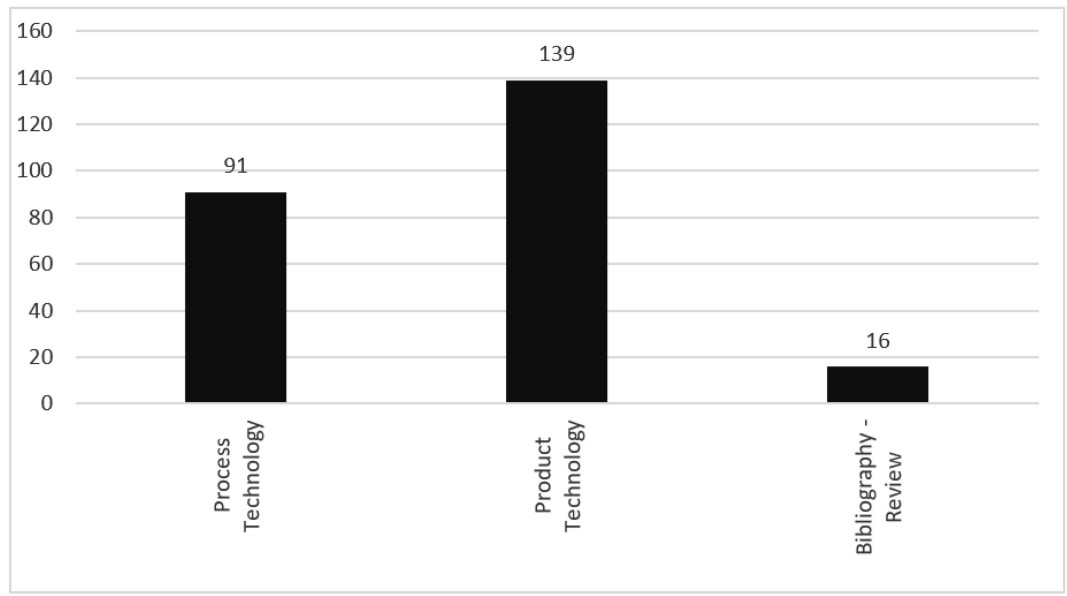

**Figure 8.** Product development by innovation projects (source: own elaboration).

It was not possible to group or classify the process technologies involved in the publications since they are varied and little recurrent in more than one publication, some considered innovative by the authors given the process employed being developed in the methodology of the work itself, as an example the work of [76] in which the integration of an IA tool was added to the design process of several different points in the control of the project.

Another aspect present in more than one publication refers to the dimensioning of units and controllers, such as the works of [118,119] in which process technologies are employed to innovate in the correct design of a motor-reducer unit; and [83] which features an innovative controller, synthesized to ensure performance improvements, robust tracking, and disturbance rejection.

Micro-electromechanical systems (MEMS) consisting of process technology used to create small devices or integrated systems that combine mechanical and electrical components are employed in the work of [120], where it is possible to apprehend that they are manufactured using integrated circuit (IC) batch processing techniques, and can vary in size from a few micrometers to millimeters. These devices (or systems) can detect, control, and act on the micro-scale and generate effects on the macro scale. And although they are not the main process technology employed in other surveys of the sample, they are mentioned and used as the basis for the development of other processes, as is the case of publications [121–123].

In this set of 91 publications in which process technologies are adopted, it can be stated that a relevant result is associated with the indication that the market can explore the results presented in most publications. Machines, equipment, devices, and even information are described to add value and achieve strategic objectives, as well as explained in the publications of [120,123–125].

In the 139 publications classified as product technologies, it is also difficult to classify the typologies given the variety of fields and applications. So, we sought to present the associations with recurrent aspects in at least four publications. Recurring standards and specifications related to composition, properties, and quality requirements are recurrent:

- Product specifications: product technologies can produce incremental and progressive changes in technology, and thus, through such changes, it is possible to produce a better product for specific markets. These are examples of [95,126–128] works.
- High quality: the aspect of the quality of the product technologies is present in the publications whose results explain vibration control, the performance of sensors, embedded systems, and vehicles, parameterization of control systems, quality of high-value components used in the aerospace industry, and vehicle safety, being the following publications, examples of the quality aspect [129–134].

In addition, since knowledge and new ways of transforming ideas into products and services, it is important to highlight an excerpt from publications that employ product technologies in the various areas that make up the classification carried out: prototypes of applications focused on piezoelectric actuators [87], robots and robotic systems with health applications, automotive, food industry, bioengineering [95,135–137]; sensors and actuators [2,138] among others. It should be noted in this context that products range from education and entertainment to design for manufacturing and assembly, with mechatronics at the heart of these product technologies as a holistic approach, with product technology being the element that ensures the emphasis on communication, integration, and collaboration.

The 17 publications of an essentially bibliographical/literature review nature that do not specifically involve product or process technologies are researches that vary from subjects such as theory and methods applied in research and education activities in the field of innovation to engineering for the development of new products in Universities [139]. For example, Ref. [140] reviews the close integration between robotics and neuroscience.

Most of these publications discuss the curricula and courses of mechatronics worldwide. Ref. [141] address the implementation and evaluation of a module on robotics in undergraduate/graduate that integrated structured and unstructured learning experiences with the collaboration of third-sector partners. Ref. [99] explains the efforts of the Technical University of Denmark (DTU) to incorporate the teaching of mechatronics into its educational program of 'Design and Innovation' and the use of a special type of Unified Modeling (UML).

Figure 9 shows the main categories of subjects, defined after analysis of the sample of publications and reviewing the state of the art for constructing the analysis framework. To exemplify each of the areas and classifications of this thesis, it was established as a parameter that the researchers to exemplify the discussions must obey the citation criterion, and the examples are always presented based on the most cited publications of the sample.

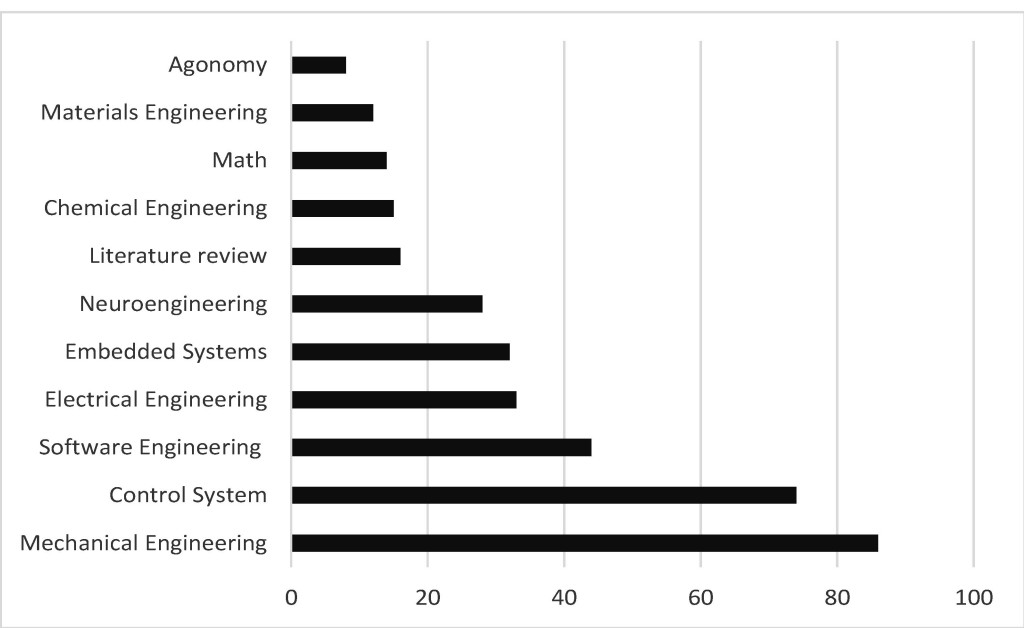

**Figure 9.** Principal subject categories (source: own elaboration).

5.3.2. Product Development by Type of Technology Used (Degree of Project Innovation)

Some areas, such as Agronomy, Chemical and Materials Engineering, and Mathematics, could be presented more expressive. In Agronomy or Agribusiness, the publications analyze new products and processes in developing horticulture and large-scale plantations, using high technology, focused on sensors, energy solutions, and roofing materials [142,143]. In the sample of publications whose centrality is agribusiness, it can be seen that autonomous agriculture is the central focus of the study, and mechatronics permeates the modeling and control of agricultural machinery by using mechatronic components and systems.

According to Figure 9 in Chemical Engineering, it is possible to observe research that employs processes and technologies from genetic improvement to the chemical composition of food. In Materials Engineering, the study of materials' bonds within mechatronic systems is observed by assembly molding as in [144,145]. The sample also presents two publications [56,146] on toxic gas control using Programmable Logic Controller (PLC) from an integrated mechatronic approach to process automation. The emphasis is on the need to control chemical release by industries and how mechatronics is crucial to it.

Mathematics is also a category of minor subject matter. It is present in descriptions of algorithms and works on robotics, as in [147] that report the first humanoid robot in space to perform experimental tasks. Ref. [75] reports the application of an innovative approach to the design and development of prosthetic hands, research in which mathematics is present as a method, and the detailed description of the calculations involved in its programming showing how the development of the prosthesis takes place from the design and calculations to the preliminary analysis of its prototype.

The research of basic nature concentrates on engineering education, with 16 publications. All of them use some method of bibliographic research in their execution and are associated with a typology of descriptive and explanatory research. For data collection, they use a survey or case study, considering that most of them are applied in some Universities or perform analysis of the curricula of mechatronics courses. The perspective of mechatronics as a science in this subject category is present through conceptual mappings and systems thinking. The work of [99] relates the effort to investigate the formulation of a new area of undergraduate concentration in Mechatronics as part of the Bachelor of Mechanical Engineering. Refs. [148–150] also work on the perspective that from globalization, outsourcing of services, and certain types of engineering work associated with technological advances, it is necessary to think about the redefinition of skills for engineering curricula

that cover various disciplines, defined in a thematically integrated curriculum. Among the 16 works in this subject category, 12 dedicate part of the results to present the difficulties associated with updating and curricular changes based on the restrictions of an existing curricular structure based on integrating areas.

In addition, it is noticed that algorithms and control systems appear associated in publications that design product approaches. For example, Ref. [151] proposes an innovative design method known as "MIRA" (Modular, Intelligent, and Real-Time Agent) to represent products and also mechatronic components in manufacturing systems. Ref. [152] reports the design process used for a medical application of a mechatronic system in a teleultrasound robot. The processes of robotic teleultrasound, as well as the prototype of tele ultrasonography and the development of its control device, are presented. Ref. [153] describes the development of an integrated model for designing and developing mechatronic products in the context of the agile production system. The study attempts to eliminate unnecessary stages of product design and simulation and increases the number of ideas tested. The experiment with hyper-tracer robots confirmed the proposed method's better consistency and agility and mentioned the cost reduction factor as an advantage. The work of [79] is presented an innovative design for a robotic dolphin to achieve high maneuverability and long durability, combining the advantages of underwater gliders.

In the subject category neuro-engineering, the publications are focused on health solutions, aiming at neurological rehabilitation, development of innovative solutions, and enhanced control strategies, such as the publications of [154] who explains the development of a unit of action in artificial hand prostheses, humanoid robotic hands, in claws. Finally, Ref. [136] presents the pilot study that documents the capacity of patients undergoing long-term robot-assisted rehabilitation treatment.

It is complex to classify research into only one area. Most tangentially and apply two, three, or even more areas of the classification established for this analysis, mainly mechanical engineering with high expressiveness and always permeating more than one theme. The research of [155] presents an innovative solution that combines gait and brake equipment in a highly integrated mechatronic system. It proposes discussing the new generation of engine tricks that combine high efficiency and low dead weight with reduced life cycle cost resulting in superior performance than conventional solutions.

Research addressing control systems, sensors, and actuators has high expressiveness in developing mechatronic products. In addition, it is noteworthy that they are always more associated with validating the performance and quality of products and applications. The most cited work on the subject is the research of [112], which addresses mechatronic systems as innovative products adopting embedded control.

Software engineering is an area that is also highlighted in the sample, predominating in more than 40 articles. Closely associated with electronics, the research is of an applied nature and exploratory character, uses the quantitative methods of modeling and simulation and qualitative case study and SSM. To exemplify this excerpt, Ref. [76] explains the development of mechatronic products from a strictly integrated methodology that incorporates an approach of continuous time energy graph systems and high-level Petri nets for the rigorous description of information in a knowledge-based design environment called Schemebuilder. Ref. [77] presents in the article a finger design that combines a reduced diversity of pieces for construction of five kinetically different fingers. And from modeling and simulation, they find that the existing hand designs used in robotics cannot yet compete with the performance of elastic elements. The strong evidence that product design is associated with cost reduction stands out.

The distribution of publications in the subject categories presented by [22] allows us to infer that regardless of the quantitative distribution of publications into categories, the classification proposed by Bradley fits very appropriately in the list of publications analyzed. It is observed that the greatest exponents of subjects, such as software engineering, mechanical engineering, and control systems, reflect the general panorama of what is possible to apprehend from the sample: we have a great exponent of works presenting

and discussing components of mechatronic systems involved in manufacturing to deliver products and new technologies to the market. The number of results is also significant that, regardless of the object of study, develop or employ integrated production systems with new technologies that rely on integrating mechatronic systems.

In addition, it is important to highlight that the distribution of publications and their applications allows us to infer that the development of mechatronics over the years has made it possible to cover issues such as biology and agronomy applied to mechatronics, studies, and products that present an analysis of human movement and interface with the nervous system, in addition to how diseases, loss of movement and limbs of the human body has been used as an object of neuroscience with associated mechatronic solutions.

5.3.3. Classification of Publications by Type of Components in a Mechatronic System

Following the analytical framework logic, the next analysis classified the publications by type of components in a mechatronic system. Remembering that the same publication may be associated with more than one component of a mechatronic system, and the chosen classification criterion is the one associated with a component predominance and the publication's object of study.

In Figure 10, sensors and instrumentation are the measurement systems used in the product to control operating and environmental conditions. Sensors are the components of the measurement system that respond to a physical parameter. They can be accompanied by transducers, components that convert a certain type of energy into another that the system can better process.

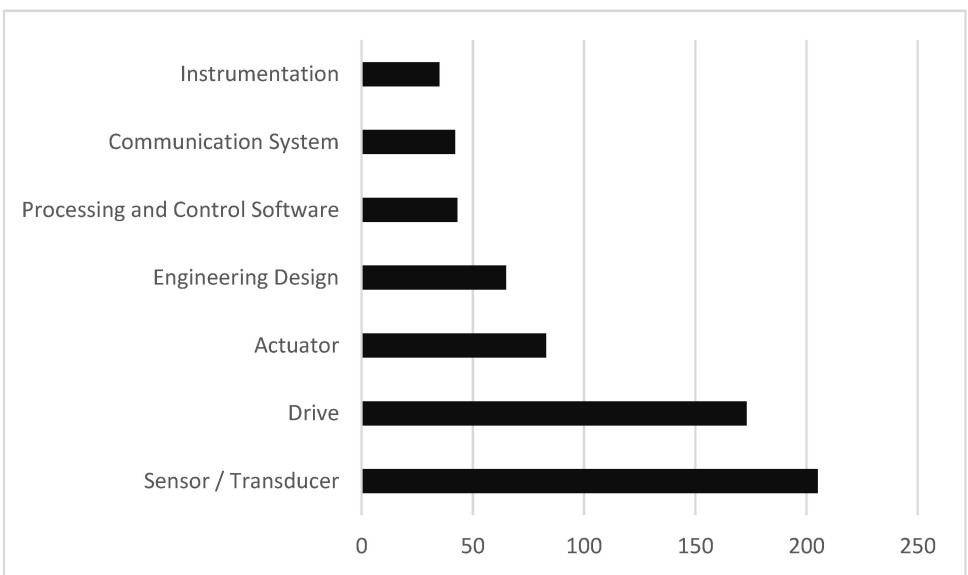

**Figure 10.** Classification of the sample publications by Mechatronic System component (source: own elaboration).

Sensors are the most present components in the sample; they are mostly associated with actuators, instrumentation, control systems, and drivers, which makes it difficult to fit the publications into only one component. Development of sensor manufacturing techniques, use of sensors to improve robot performance, and automation [156–159]. Ref. [147] report the development of the second generation Robonaut, Robonaut 2 or R2, scheduled to reach the International Space Station in early 2011, where the increase in the force sensor contributed to the improvements and changes in one of the most valuable tools for astronauts. Ref. [135] report the deployment of sensors in flexible automation for assembly on the go.

Drivers also have diverse applications when looking at publications classified in the category; they are always associated with software engineering, sensors, and actuators. In software engineering, they repeatedly address the variety of mechatronic systems operating

under electronic and computerized control and the associated drivers. Information Technology (IT) in the context of mechatronics includes computers and digital signal processors (DSPs), which store and process information, communications, and the Internet, which transmit information, as well as various computer-aided design (CAD) software packages. To illustrate, some publications associated with mechatronics in the automotive sector apply drivers and control systems to the context of the automotive industry and electric vehicles [47,159–162].

Drivers are also prominently identified in publications on mechatronic correction for sliding angles, steering systems, and cooperative redundant drivers [128]. In addition to being developed with the goal of accuracy improvement of robot drives [163].

Actuators are robust components used to correct the operation of the system. They are mechanical or electro-electronic solutions that act directly on the mechanism that performs the basic operation of the product. Drives are electronic circuits that interface between the control signals generated by the microprocessor system and the part of power responsible for supplying power to the actuators. The sample of publications analyzed presents actuators predominantly associated with control systems [164–166] and drivers [167–170].

Among the most cited in which actuators are a central element, it stands out [171]. It explains using an electromechanical valve actuator (EMVA) formed by magnets and balanced springs as a promising tool to implement innovative motor management strategies. Ref. [159] discusses two mechatronic innovations in hard disk magnetic actuator systems to sustain the 60% annual increase in the storage density of these devices. The first uses bandwidth two-stage actuator systems to improve the accuracy and trackability of the control system's read/write head. And the second is the instrumentation of disc drive suspensions to improve the suppression of airflow-induced suspension vibration in hard disk drives. Another is [87], who discusses the concrete characterization of piezoelectric actuator elements of great displacement. The article demonstrates the promising potential inherent in piezoelectric actuators for innovative mechatronic system solutions through two examples. It presents an efficient construction method for high-performance piezo actuators with high stroke capacity. Notably, the integration and the role associated with mechatronics as an element for cost reduction and implementation of innovation as a differential attributed to the characterization of the products.

Engineering design is the engineering solution for the purpose that the product must meet. When integrated into mechatronic equipment, the basic design must consider the other mechatronic components from its conception. Unlike the traditional approach, this solution must observe the requirements of the sensors and actuators to be used, and these usually influence the choice of the microprocessor to be used in the project. The publications emphasizing this component of a mechatronic system are in smaller numbers, just over fifty being the most cited those of [172–174], and the example of the higher incidence of the approach of engineering designs in the sample publications are the methods employed in them. As in [175], who use engineering design to adopt a new learning process to train real-life tasks, propose a system to challenge older people to solve new motor problems in real-time, inducing variable environments that need an active response.

It is observed in the engineering design applications of the sample that there is a tendency to change an emphasis on problem-solving projects and design. What [176] suggested eleven years ago is one of the five main changes in twenty-first-century engineering education, with design being a very important component distinguishing engineering from other areas such as applied physics.

The instrumentation components, communication systems, and processing and control software are the least expressive. They appear in less than forty publications. Instrumentation is related to signal treatment, interfacing it with the microprocessor control system and reducing noise from the measurement environment, as in the publications [177–179]. One of the most cited publications [107] proposes an intelligent and innovative switch for automotive applications that must be instrumented to manage communication with the prototype of the proposed solution.

Communication systems are physical or electromagnetic means through which the signals are produced and used by sensors, microprocessors, and actuators transit. Signals, communications, and control in the sample of analyzed publications involve aspects related to the collection, analysis, and processing of continuous and digital signals; and data communication techniques and technologies. That is, it is perceived that communication systems, by the very nature of the component, involve the entire structure of a mechatronic system; being difficult to find publications that were strictly applied to communication systems and that did not touch two, three or even more components of a mechatronic system, not being the emphasis the communication system.

Because they permeate the conception and structure of a mechatronic system, it is not possible to observe a niche of specific applications, so among the most cited publications, we exemplify those that portray in the contingent of those that emphasize communication systems, as they present themselves: (i) the question of speed, capacity, and intensity that software engineering applications employ in communication systems is quite present, as in [122] who present the tutorial on the communication of high-speed vehicular data in high voltage lines. The paper reports a communication system between different components of the vehicle, using the powerline as a communication channel without separate data lines; (ii) the issue of interface and economy in terms of physical infrastructure, as in [180] who present analyses and investigations of a new communication system to reduce interfaces and save cables for self-sustaining mechatronic modules, resulting in new concepts of modular machines; and others (iii) emphasize development of prototypes of communication systems and approach to the design of the software, considering and discussing the technologies used in the design core. Finally, some articles discuss the associated physical components of communication systems, as in [181–183].

Processing/control software, on the other hand, are the components that store and control the main functions of a mechatronic product, being considered the main logical component of the system. The sensors and transducers emit the information that is processed, and the routines are executed in a way that commands the operation of the product. They are also elements that are difficult to classify/identify as individuals without being associated with a methodology section of the publication or as a practical result of the development of some mechatronic product. Ref. [184] presents a new semi-active intelligent control technique for vibration replacement using rheological magnet damper (MR) effects through processing and control software. And [185] shows the software control used in the crossover robot remote control competition design.

### 5.3.4. Product Development by Degree of Innovation

Still, in the case of the product development variable, the publications were classified according to the degree of innovation of the product design effort (Figure 11).

Figure 11 shows that most publications, 148 (one hundred and forty-eight), deal with incremental innovations. That is, they employ and describe innovations that improve products and processes, reduce costs, and perform the same process or do the same function in a product differently when compared to others already existing. Ref. [186] has already discussed the expressive presence of incremental degrees of innovations; their study shows that incremental-type technologies seek the best form and methods for creating a product. Ref. [187] depending on the product, the specifics and final features can add or reduce steps in the development process.

Ref. [188] investigated 72 companies, and of these, 46 had innovation projects. The study reports that companies have strong processes using incremental innovation with the goal of improving their products to sustain a certain competitive position in the market, associated with cost, product and brand maintenance, customer maintenance, and business strategies.

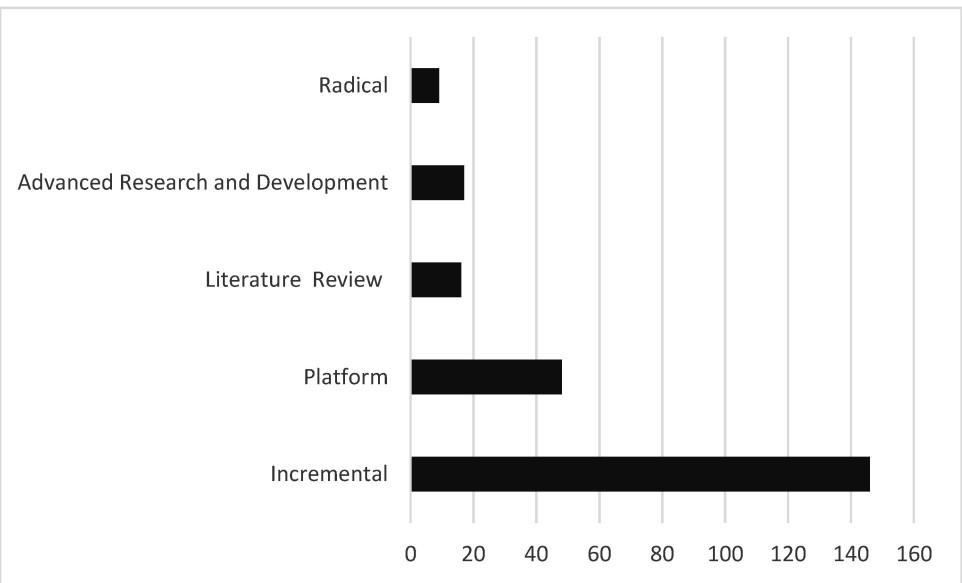

**Figure 11.** Product development by degree of innovation (source: own elaboration).

The literature reports that traditional product development models do not adhere to types of projects with a high degree of uncertainty and complexity, which are typical of radical innovation that involves new technological advances and/or new markets. For [189], higher levels of innovation present a better performance in the introduction of new products, but the associated complexity already characterizes the use of such technologies as being in smaller number, quantity or scale. This aspect may explain the low number of radical innovations reported in the literature compared to the incremental ones, as already mentioned, which suggests a hypothesis for future research.

This brief context of the research of [154,155,190–194] translates applications and important characteristics found in the publications classified as to the types of technology employed in the sample.

The degree of innovation of the incremental type, represent the most expressive contingent in the publications of the analyzed sample, comprising 146 publications. In the conceptual exposition of the analysis framework built to define the scope of this research, it is described that this type of incremental technology is associated with improvements in productivity, performance, and any attribute that through continuous, small and incremental innovations are incorporated into products and processes.

There are numerous applications and associations of incremental degree in the sample, some of the most recurrent are associated with: changes in sensors and actuators in heart rate monitoring [195], new sensors to improve performance of robotic or neurorehabilitation prostheses [137,156,157,196]; incremental innovations associated with algorithms and artificial intelligence applications for mechatronic products [158,197–201]; incremental innovations applied to actuators, sensors, and instrumentation to improve performance of mechatronic products in areas such as healthcare, automotive, Refs. [81,126,159] application of robots in service tasks (personal assistance, education, social tasks) in which incremental innovations are employed to improve parameters such as weight, kinematic configuration, the layout of masses and structures [75–77,202] incremental innovations in communication systems between different vehicle components [122,125,203,204].

It was possible to identify that platform-type technologies are present in 48 publications. The context of the results of these publications are of technologies considered as the advancement or next generation of an existing one, always associated with the development processes that incorporate significant innovations in the product and/or process, and mainly within the set of publications analyzed, generating a new family of products. Among the most cited publications in the sample are: families of controllers [96,205] and sensors [80]. Some of the publications whose technology results in a new family of products,

have an interesting feature associated with the controllers that always appear linked to the previous generation, and the new generation is characterized as a platform having its origin due to efforts to increase performance and incorporate new communication protocols. The works of [190,205–207] are expressions of this finding.

Another common aspect, in publications classified according to the degree of platform-type innovation, refers to interconnectivity and processing speed, in which the same infrastructure resources are used to offer different products and services, thus reflecting a greater degree of innovative capacity for the next generation of products and/or services, such as the works of [81,93,175,208].

The degree of radical innovation was found in 9 publications, reinforcing what [188,189] point out about innovation management being closely linked to uncertainty management. The better the management of uncertainties and risks, the greater the chances of ensuring the project's success to be executed. That is, it is verified that the scope of the projects is focused on applications whose associated technologies are mostly incremental. Two publications classified as radical in the sample address aspects of disruption when discussing the introduction of macro technological and market discontinuities. Refs. [106,109] developed an innovative anthropomorphic robotic hand called DEXMART, radical innovation since, unlike the others, it has proven the introduction of suitable design solutions aimed at simplifying and reducing costs, robustness, and performance. Ref. [209] present a telerobotic haptic system for stereotactic neurosurgery, with tests that have already proven accuracy, safety, reliability, and ease of use, thus proving that the system meets all the requirements for effective clinical use.

What can be seen in the nine studies classified as radical [2,106,209–214] is that the greater the degree of uncertainty associated with projects, the more variables and, obstacles, problems are observed in the dynamics and characteristics of the creative process involved. The results show that such projects are part of contexts of knowledge that we can call less stable and subject to internal environments of greater autonomy and less predictability.

The number of research classified as Advanced Research and Development is 17. In this classification, the development of technological knowledge for future applications is not necessarily associated with commercial purposes. It is not considered an innovation project precisely because of the characteristic that the result could be an innovative product in the market. They are publications strongly associated with renowned international research centers, Universities, and projects funded by large research institutions. Examples: the publication on a robot sent into space and its constitution [147], the conservation and planning project of underwater archaeological artifacts [215], and publications on advanced research and development programs such as the "Mechatronics Program" and "Applications of intelligent learning systems" in Finland that demonstrate the joint project efforts between industry and research centers [216].

On the other hand, the theoretical research that applies bibliographic review focuses on engineering education and is essentially theoretical (16 publications). All have basic nature and as methods use surveys or case studies, and are associated with Universities, Graduate Programs, or analyze curricula of undergraduate courses: [98,139,141,149,216–218]. The perspective of mechatronics as a science is present through conceptual mappings and systems thinking. The work of [99] relates the effort to investigate the formulation of a new area of undergraduate concentration in Mechatronics as part of the Bachelor of Mechanical Engineering. From globalization, outsourcing of services, and certain engineering work associated with technological advances, it is proposed to redefine skills for engineering curricula that span various disciplines, defined in a thematically integrated curriculum. The final document describes the effort, including the difficulties associated with working within the constraints of an existing curriculum structure at a small engineering college.

Despite presenting several characteristics in common, innovative projects still need to form a homogeneous class and, among the possibilities that exist outside, employ the one that differentiates their degree of innovation [66]. It was possible to verify that as to the degree of innovation of the project and type of technology used in the sample, the

publications employ/develop incremental innovations, followed by innovations of the platform type, having a small number of radical innovations.

In the sample of the degree of innovation associated with products and processes, it is observed that the improvements in productivity, performance, precise execution, and design change offered by incremental innovations are the main reasons why small and incremental changes over time are executed in the objects of study, regardless of the area of application. These incremental innovations are tied to cost, time, and feasibility, exceeding the performance obtained by radical innovations.

Old but relevant publications such as [219,220] indicate that continuous improvement was widely practiced with good results in Western factories during World War II. And from this experience, attention has turned to more radical innovations in products and processes within manufacturing; however, while it is certainly true that more radical innovations have enormous potential to improve productivity, increase flexibility, and raise quality standards, among other aspects, it is also increasingly clear that obtaining benefits of this type is by no means a simple or automatic process.

It is important to note that in the sample publications, regardless of the technology employed, time is an essential aspect, a key element in analyzing the innovation made possible by technology. Organizations, companies, and researchers invest time improving existing products (incremental innovation) or developing new products that seek to break down barriers and create new markets (radical innovation) and save or gain time through the adoption of processes, methodologies, and artificial intelligence are strengths found in the publications [74,221].

It is noticed that the publications of the sample that predominate the incremental innovations and platform reinforce the studies on the degree of innovation of the projects, recognizing the influence that the structure of the product itself exerts on the characterization of the knowledge involved in its development process. And we can conclude that the publications on mechatronics analyzed in this research, regarding the degree of innovation, predominantly employ incremental innovations in the various areas of application of mechatronics, as previously exposed; followed by platform-type innovations, more restricted to the health area and the automotive industry, with radical innovations being the minority of publications.

## 6. Discussion

In general, the bibliometric analysis showed that innovation is essentially an attribute of mechatronics, present in all articles in the sample. Innovation also appears essential in terms of innovation in the curriculum, discussions about terminology, or even in reporting the development of devices and technologies.

The set of methods employed was intended to achieve the proposed objective by glimpsing through the results of the analyses, the academic impact of the analyses, the collaboration network of the publications, and the focus of each of the methods used in the analyses, to subsidize responses to the mapping of how it is developing the scientific of mechatronics. Quantitative bibliometric analyses were performed to gather more objective evidence to support prediction questions.

In the qualitative analysis, the content of the articles was analyzed to identify their correlations, in addition to understanding the content and the construction of knowledge through a systematic review of the literature.

It was demonstrated through the application of Table 2 (Analysis Framework) and the results of Figures 4, 5 and 8–10 that the scope of the sample's research has the degree of innovation and technology associated with the object of study, predominantly associated with mechatronic systems, mechatronic products, robotics, manufacturing, engineering education, curricula, product design, machinery, and product development.

The articles were also classified according to the components of a mechatronic system, which allowed us to verify most studies using sensors and instrumentation. Recurrent in more than half of the sample, actuators and drives are the components that appear in

second place in the sample, whether in the description of characteristics of products and mechatronic systems both in terms of functional, mechanical, and electronic interaction and regarding discussions between the information technologies employed.

It is important to highlight that the 4th Industrial Revolution, which has a greater focus on the manufacturing area, is also present in the context of articles from the last ten years. The analysis of publications by country over time shows dissemination of the distribution of the theme across different countries, not only in those that are at a higher stage of technological development, such as Germany, Italy, the United States, and China, with the expressiveness of publications evidenced in Figure 5.

## 7. Conclusions

### 7.1. Concluding Comments

The main findings, questions, and analyses about the scientific constitution of mechatronics with innovation as an attribute are summarized in Table 4.

**Table 4.** Summary of Conclusions (source: own elaboration).

| Mechatronics | | |
| --- | --- | --- |
| **Innovation as an Attribute/Asset** | | |
| The bibliometric analysis employed in 646 publications | Qualitative Analyzes employed in 246 publications | Framework built for classification of publications according to the analysis variables characterized for the study |
| **Components in a Mechatronic System** | **Distribution by Predominant Country** | **Qualitative Systematic Review** |
| Sensors, instrumentation, and actuators predominated. | Germany, Italy, the United States, and China. | The expressive Robotics theme in the sample. |
| **5 Classified Scenarios** | | |
| (1) practical approaches directed to the development of products, mechatronic systems, robots, and automation; <br>(2) researchers that study engineering curricula and education; <br>(3) studies involving components of a mechatronic system; <br>(4) the employment of artificial intelligence, engineering design, and product/machine design; <br>(5) methodologies for the design of mechatronics. | | |

First, the finding that there are few theoretical studies on the constitution of mechatronics as a science; much has been discussed about conceptual expansion or reductionism, about "what is mechatronics", and "how mechatronics can be defined" over the years, but little has been discussed about the scope, types of research and what has been produced in the area. Along with technological changes and mainly associated with innovation, mechatronics has been evolving with new application opportunities and challenges in terms of field of study and research. It is possible to point out that the use of the term innovation associated with mechatronics in most publications goes beyond the operational level, characterizing the attribution to the term always associated with the application, developments, and prospects that product, design, robot, or system may provide for the market or future research. At the same time, it appears that the result of many publications, when reinforcing innovation, associates it with return on investment, operating costs and highlight the advantages of technologies employed for commercial purposes.

The second important point concerns the observed scenarios. It was possible to verify that the interdisciplinary work between mechanics and electronics is very important and permeates almost all research, without the need for standardization or definition of a special technical term; research projects and product development can be easily placed at an interface between mechanics and electronics. For this kind of technology, the literature have pointed product development reference models, such as [222,223] for guiding companies to reach higher technical performances at low costs and time-to-market.

The results and conclusions of the articles show that innovation plays one of the main roles in the mechatronics field, most likely due to the multidisciplinary integration that

the scope of innovation in product engineering is providing nowadays. Some scenarios can be classified into: (1) practical approaches directed to the development of products, mechatronic systems, robots and automation are always directed to meticulous and detailed analyzes of case studies with applications directed to the automobile, satellite and aerospace market; (2) researches that study engineering curricula and education discuss a lot the standardization of curricular approaches and components and their impacts on academic or market direction; (3) in studies involving components of a mechatronic system, they are more associated with validation of performance and quality of products and applications; (4) the employment of artificial intelligence, engineering design and product/machine design, although very diversified in the sample, are mostly associated with cost reduction; (5) the scenario of methodologies for the design of mechatronic systems appears with contributions that place mechatronics in an evolutionary stage involving systemic thinking, project management and conceptual mapping.

The qualitative systematic review and classifications from the analysis framework also bring insights into robotics. A significant portion of the studies is dedicated to technology and robots for application in automation, health, in the aerospace and manufacturing context. Even in different contexts, it is clear that the development of robotic technology is very strong in the literature and carries technical aspects ranging from the proposition of algorithms to innovation in mechatronic devices, computer simulation, sensors, drivers and mechatronic systems. Very meticulous and specific innovations have been proposed in the field of robotics, confirming the area as one of the main pillars of the scientific constitution of mechatronics.

In general, the bibliometric analysis carried out showed that innovation is essentially an attribute of mechatronics, present in all articles in the sample. In terms of innovation in the curriculum, discussions about terminology or even in reporting the development of devices and technologies, innovation appears as an essential attribute.

As previously exposed, Refs. [21,22] discusses questions that he considers unanswered for the evolution of mechatronics and one of the many discussions is related to mechatronics, if this remains an area that should, or should not remain separate and distinct from other engineering approaches and engineering project. Or are proposals, studies, and advances enough to make it incorporated into conventional engineering? Our analysis so far understands that the constitution of mechatronics based on interdisciplinarity is essential for the characterization of a science that is not only mechanical, not only electrical or software, its correlated areas alone, do not contemplate the complexity of what by nature it is interdisciplinary. So far, it has also been shown that innovation is a strong element, present in the historical and evolutionary chain of mechatronics. This becomes clear and evident when the analyzed publications always have the term innovation (regardless of type and degree) associated with mechatronics. Besides a small sample of advanced research and strictly theoretical approaches, the large majority of the analysed articles presented innovation projects, but most of them classified as incremental innovation. Platform innovation also represents a considerable number of cases, but radical innovation is present on a small percentage. It can be seen as an interesting issue, once mechatronics has begun as incremental innovation by inserting electronic and software features on mechanical-bases devices. In terms of innovation in the curriculum or discussions about terminology, innovation figures as an important attribute as well.

The conception of mechatronics as a scientific field is evidenced and perceived in different forms and symbolic productions within the very components that were analyzed in the analysis framework. The system of relations established with the areas that make up mechatronics is the basic premise of its existence, there is no way to observe sensors, actuators, communication systems, instrumentation, control, without them permeating and being associated with mechanics, electronics, or software. That is, it can be inferred and stated from the results already presented that mechatronics as a science is constituted by means of all the apparatus of theories, laws and specific forms within the interdisciplinarity that occurs between mechanical, electronic and software engineering, within of the insepa-

rable attributes that it carries in technological applications that, through innovations, take shape in the structure of all application components for mechatronic products.

From this context and all the aspects raised by psychology, information science, pedagogy and psychopedagogy, it can be seen that for the construction and consolidation of a concept it is necessary more than the establishment and scientific dissemination, it is necessary to understand the relationships, psychological instruments, conceptions and context of the subject, scenarios, epistemological conceptions, among other elements that contribute to the break with common sense, allowing to tread a path for an area to consolidate itself as a science.

The qualitative analysis from the analytical framework also brings insights into robotics. A significant portion of the studies is dedicated to technology and robots for application in automation, health, aerospace, and manufacturing. Even in different contexts, it is clear that robotics development is strong in the literature and carries technical aspects ranging from the proposition of algorithms to innovation in mechatronic devices, computer simulation, sensors, drivers, and mechatronic systems. Very meticulous and specific innovations have been proposed in robotics, confirming the area as one of the main pillars of the scientific constitution of mechatronics.

### 7.2. Limitations and Future Work

As for the limitations, some points can be mentioned:

- It is a challenge to differentiate mechatronics from concepts from the lateral areas such as, for example, Industry 4.0, which can provide similar results.
- Future work to advance the field, which is nowadays scarce, must be expanded.
- Scope of publications—working only with articles could offer a more accurate sample related to the publication type. It would also offer the possibility of performing more in-depth bibliometric analyses. We analyzed the area as a whole without restricting the publication type and outlined a general overview. However, we could not go into details and carry out more specific analyses given the high quantity and diverse nature of the publications.
- High number of publications—during the analysis, other levels and variables of analysis were observed regarding the stratification of subjects and attributes of software used, among other possibilities. However, any change requires reanalysis and reading the 246 publications of the qualitative analysis, that was considered not necessary to the scope of this paper.
- Frameworks for qualitative analysis on the subject are little explored—it was difficult to identify qualitative analysis frameworks in the literature to help and support the analysis.

Despite the limitations, the study is scientifically useful since it contributes to constructing the scientific field that characterizes mechatronics as a science that drives innovation. The contributions highlighted here should be understood as the results of a general analysis of studies in innovation and mechatronics over the years to gather what has been produced and is available in the main databases in the world.

The analysis presented here can be used as a proposition for future studies, which could funnel established classifications into smaller contexts for more detailed analyses.

**Supplementary Materials:** The following supporting information can be downloaded at: https://www.mdpi.com/article/10.3390/asi6040072/s1.

**Author Contributions:** Conceptualization, A.C.C.F. and S.C.M.B.; methodology, S.C.M.B.; software A.C.C.F.; formal analysis, A.C.C.F. and S.C.M.B.; data curation, A.C.C.F.; writing—preparation of the original draft, A.C.C.F. and S.C.M.B.; writing—proofreading and editing, A.C.C.F. and S.C.M.B.; supervision, S.C.M.B. All authors have read and agreed to the published version of the manuscript.

**Funding:** The APC was funded by the University of Brasília—UNB.

**Data Availability Statement:** All data is available in the article itself.

**Conflicts of Interest:** The authors declare no conflict of interest.

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
