# Peer review of "Mechatronics: A Study on Its Scientific Constitution and Association with Innovative Products"

_asi, doi:10.3390/asi6040072_

Round 1
Reviewer 1 Report
This paper presents a bibliometric and qualitative analysis of mechatronics literature. The review is presented as follows:
The abstract is confusing, with long sentences (e.g., the second one). It was important to state the main conclusions like the 5 scenarios briefly.
English revision is necessary.
Introduction:
When defining mechatronics, please provide a reference because there are already definitions in the literature.
Third paragraph. I do not understand the first sentence.
It is not clear why data science is mentioned. The focus of the authors’ work seems to be innovation via mechatronics. A clear research question seems to be missing. It was also important to understand if there is related work on the topic.
Section 2
The methodology flow is well explained
Stage 3 – the first paragraph is confusing.
Very long paragraphs after Table 2 does not allow a fluid reading. It was important to reduce the text.
Section 3 – Results
A revision is necessary to remove duplications and focus on the results. For example, the aspects of php script were already mentioned. Here the authors should be specific about the results.
Table 3 is a good output. An extra column with a summary of the study could be included.
Figure 4 quality is insufficient.
Figure 5- please adjust VOSViewer parameters to allow better visualization of the network
Graph 6 – some of the items are not technologies
One of the main problems in this study is the long list of paragraphs, some of them unrelated - which could be improved in the revision. Therefore, it was interesting to include at the end of the paper a table summarizing the contribution of mechatronics to innovation (or in an appendix if it is too extensive). A discussion section (using part of the long conclusions) is a possibility.
Section 4.2. the first sentence seems strange – the sample already mixed innovation in the keywords, so it is not surprising. The claim is unfounded. The second paragraph is also confusing and not part of the limitations.
I suggest adding another limitation – it is challenging to differentiate mechatronics from other concepts in this study. For example, Industry 4.0 could provide very similar results.
Future work to advance the field is scarce and needs to be expanded.
Very long paragraphs make the paper more difficult to follow. An English revision could improve paper readability.
Author Response
Response to Reviewer 1 Comments
This paper presents a bibliometric and qualitative analysis of mechatronics literature. The review is presented as follows:
The abstract is confusing, with long sentences (e.g., the second one). It was important to state the main conclusions like the 5 scenarios briefly. English revision is necessary.
Response: The abstract was revised and the scenarios included as results in the text. We revise English as requested
- Introduction: When defining mechatronics, please provide a reference because there are already definitions in the literature. Third paragraph. I do not understand the first sentence.
Response: We have inserted more references as to the definition of mechatronics and removed the third paragraph, rearranging the text.
- It is not clear why data science is mentioned. The focus of the authors’ work seems to be innovation via mechatronics. A clear research question seems to be missing. It was also important to understand if there is related work on the topic.
Response: The analyses carried out through scientific collaboration, have the core of their emergence in the studies that investigate science by science itself, that is, data science. The paragraph has been rewritten to improve understanding.
- Section 2 The methodology flow is well explained
- Stage 3 – the first paragraph is confusing.
Response: Text rearranged, paragraph deleted.
- Very long paragraphs after Table 2 does not allow a fluid reading. It was important to reduce the text.
Response: We have reduced the paragraphs and text as much as possible.
- Section 3 – Results A revision is necessary to remove duplications and focus on the results. For example, the aspects of php script were already mentioned. Here the authors should be specific about the results.
Response: We have reduced the paragraphs and all the text as much as possible. However, one of the evaluators requested the inclusion of the literature review sections, which occupied the space of the reduction performed
- Table 3 is a good output. An extra column with a summary of the study could be included.
Response: Publication summary columm added to table as per request
- Figure 4 quality is insufficient.
Response: Figure 4 really didn't get an excellent quality because it was only possible to extract it in a dimension that fits the ruler of the article, performing reduction of dimensions visuaus in vosviewer. As we could not improve the quality, we adjusted the text to explain the implications that this network represents, and excluded the figure, reorganizing the results.
- Figure 5- please adjust VOSViewer parameters to allow better visualization of the network
Response: adjusted parameters, if we decrease for 2, unconnected nodes they appear, if we increase, significant nodes disappear, we improve the visualization parameters.
- Graph 6 – some of the items are not technologies
Response: Terminology changed to level of Innovation to make understanding easier.
- One of the main problems in this study is the long list of paragraphs, some of them unrelated - which could be improved in the revision. Therefore, it was interesting to include at the end of the paper a table summarizing the contribution of mechatronics to innovation (or in an appendix if it is too extensive). A discussion section (using part of the long conclusions) is a possibility.
Response: The table with the summary of the conclusions was inserted, the text divided into discussion and conclusions, as requested. And we have reduced the paragraphs and the entire text as much as possible. However, one of the evaluators requested the inclusion of the literature review sections, which occupied the space of the reduction performed .
Section 4.2. the first sentence seems strange – the sample already mixed innovation in the keywords, so it is not surprising. The claim is unfounded. The second paragraph is also confusing and not part of the limitations.
Response: Innovation is part of the key terms, hence the mention in the text. We have rearranged the writing of this excerpt and deleted the second paragraph as requested.
I suggest adding another limitation – it is challenging to differentiate mechatronics from other concepts in this study. For example, Industry 4.0 could provide very similar results. Future work to advance the field is scarce and needs to be expanded.
Response: We include the suggestions of limitations in the text and take the opportunity to thank the valuable contributions of this evaluator.

Reviewer 2 Report
Dear authors,
Thanks for giving me the possibility to review the paper. It is a literature review and the field is interesting. I think it can be a milestone for future.
give a look to format (some title in a page and the paper in an other, probably at the end with editing everything will go ok).
Author Response
Response to Reviewer 2 Comments
Thanks for giving me the possibility to review the paper. It is a literature review and the field is interesting. I think it can be a milestone for future. Give a look to format (some title in a page and the paper in an other, probably at the end with editing everything will go ok).
Response: Thank you for encouragement words also for evaluating our project , we really appreciatte it. We have revised format and reference as per your indication.

Reviewer 3 Report
I have reviewed the paper entitled “Mechatronics: a study on its scientific constitution and association with innovative products”. I found the topic interesting, however, it has some caveats that need to be addressed before it is ready for publication.
The paper does not present an explicit objective or research question, so it is difficult to understand what the purpose and contribution of the paper are. Moreover, there is no literature review of previous works that would give a framework. While bibliometric studies are used to evaluate publication activity of fields of knowledge, it is necessary that they explicitly define a a clear and meaningful research question. That in this paper, it is lacking. Authors mention “prediction question” but it is unclear what it is.
Other concern is the design and base of knowledge of the study. Looking for publications using the “innovation and mechatronics or other 2 forms of the almost same noun is leaving outside a lot of research related to the topic. For example, robotics, automation, electromechanics, etc. Concluding that innovation is related to mechatronics is not a valid conclusion since it is what authors looked for. There is no contrafactual o individual analysis, all the analysis is the intersection of the terms.
The theoretical framework either of innovation or mechatronics is also very narrow. OECD provides adequate definitions of the types of innovation. For example, project, process, and product innovation have different characterizations. The same happens with “degree of innovation”.
In terms of the data, the analysis seems also incomplete and inaccurate. For example, the paper states the “country of publication”. Does that refer to the country of the journal or of the authors? How do authors deal with collaboration? Most of the papers mentioned are “et al.”, thus, it is necessary that authors consider collaboration. Also, the name of the authors seems incorrect. Bradley d (sic) and Bradley da (sic), is the same author (David Allan Bradley), but he has publications as Bradley D, Bradley DA, and other forms. This is one of the difficulties of bibliometric analysis. This paper does not take into consideration this aspect in the analysis that shows.
The authors also conclude “evidence of multidisciplinary” but this aspect also has very broad features that also need to be address properly.
The conclusions need to be revisited accordingly with the improvements of the paper.
The references are not cited properly. The articles and proceedings of the analysis are not considered references, they must be considered as supporting material or annex.
English also needs to be revised.
English needs to be revised.
Author Response
Response to Reviewer 3 Comments
- I have reviewed the paper entitled “Mechatronics: a study on its scientific constitution and association with innovative products”. I found the topic interesting, however, it has some caveats that need to be addressed before it is ready for publication. The paper does not present an explicit objective or research question, so it is difficult to understand what the purpose and contribution of the paper are.
Response: We explicitly insert the research problem and general objective in the instroduction.
- Moreover, there is no literature review of previous works that would give a framework.
Response: Due to the volume of results achieved by our research and because we observed publications of the same nature that focused on the results without exposing a review, in the same journal, we had deleted it from the text. Now we insert as requested.
- While bibliometric studies are used to evaluate publication activity of fields of knowledge, it is necessary that they explicitly define a a clear and meaningful research question. That in this paper, it is lacking. Authors mention “prediction question” but it is unclear what it is.
Response: Research problem explicitly inserted in the introduction
- Other concern is the design and base of knowledge of the study. Looking for publications using the “innovation and mechatronics or other 2 forms of the almost same noun is leaving outside a lot of research related to the topic. For example, robotics, automation, electromechanics, etc. Concluding that innovation is related to mechatronics is not a valid conclusion since it is what authors looked for. There is no contrafactual o individual analysis, all the analysis is the intersection of the terms.
Response: This article clipping is part of a doctoral thesis, which assumes innovation as an attribute, an asset in mechatronics. If we perform a search only with the term mechatronics the results exceed 33 thousand publications, which makes a qualitative analysis impossible. So as we need to establish feasible parameters for scientific research and it is observed in the context of conceptualization of innovation that its degrees and applications, have frequent recurrence in mechatronics, as well as other terms such as industry 4.0, sustainability could be associated with mechatronics for a similar study, the choice for innovation was made. In order to make this clear and explicit, we have included this inclusion criterion in the introduction.
- The theoretical framework either of innovation or mechatronics is also very narrow. OECD provides adequate definitions of the types of innovation. For example, project, process, and product innovation have different characterizations. The same happens with “degree of innovation”.
Response: We believe that the inclusion of the literature review section will solve this . The level of innovation corresponds to the variables raised in the analysis framework and were thus attributed by Wheelwright and Clark (1992), by type of technology used (degree of project innovation), that are: Incremental, Platform, Radical.
- In terms of the data, the analysis seems also incomplete and inaccurate. For example, the paper states the “country of publication”. Does that refer to the country of the journal or of the authors? How do authors deal with collaboration? Most of the papers mentioned are “et al.”, thus, it is necessary that authors consider collaboration. Also, the name of the authors seems incorrect. Bradley d (sic) and Bradley da (sic), is the same author (David Allan Bradley), but he has publications as Bradley D, Bradley DA, and other forms. This is one of the difficulties of bibliometric analysis. This paper does not take into consideration this aspect in the analysis that shows.
Response: We include in the text the specification of the origin of the data for classification of publications by country, as well as detail the author entry for different occurrences for the same name, including there is a homonym, whose publications are not part of this study.
- The authors also conclude “evidence of multidisciplinary” but this aspect also has very broad features that also need to be address properly.
Response: We made the correction to let it explicit that interdisciplinarity is an intrinsic characteristic of the area, since mechatronics only exists, because mechanical, electrical and software at the moment in which they join their practical applications, have created a new interdisciplinary area that has at the center of its essence, the indispensable interdisciplinary presence of the areas that have constricted it. We've modified the text to make it clearer.
- The conclusions need to be revisited accordingly with the improvements of the paper.
Response: We revised the text and even divided the conclusions into discussion and conclusion, to meet another evaluator.
- The references are not cited properly.
Response: We reviewed the references according to the journal's template.
- The articles and proceedings of the analysis are not considered references, they must be considered as supporting material or annex.
Response: We added a table with the publications that were part of the templates, both bibliometric analysis and qualitative analysis
- English also needs to be revised.
Response : We revised the language

Round 2
Reviewer 3 Report
We explicitly insert the research problem and general objective in the instroduction.
I saw the insert of the research problem and the central objective, however, still it is not clear what the contribution of the research is. In other words, why is important that the scientific community knows about “how the elements that constitute the field of knowledge that subsidizes the construction of mechatronics are treated from the perspective of their association with innovation? And the central objective of this article is to analyze the construction of the field of scientific knowledge that has innovation as an attribute of mechatronics, in addition to identifying the main elements of mechatronics from quantitative bibliometric analysis and scientific collaboration networks and analyzing the publications of the sample from an analysis framework”. Even more, authors need to explicitly say what is the analysis framework.
The explanation about David Allam Brandley is very confusing and hard to follow. Authors state “Note that David Allan Bradley has two distinct forms of name entry: Bradley D. and Bradley D.A. As extracted from Scopus and Web of Science, your two author entries are valid and used to designate you in publications, so there are two clusters associated with the respective entries.” Who is your two authors and designate you? The next paragraph is also very confusing. Are the authors taking about two different authors or not?
The references are still not cited properly.It is not about the format, it is that the articles and proceedings that were part of the analysis are not considered as part of references, they must be considered as supporting material or annex.
English also needs to be revised. There are parts that are confusing and hard to follow.
English need to be revised
Author Response
Response to Reviewer 3 Comments – Round 2
- I saw the insert of the research problem and the central objective, however, still it is not clear what the contribution of the research is. In other words, why is important that the scientific community knows about “how the elements that constitute the field of knowledge that subsidizes the construction of mechatronics are treated from the perspective of their association with innovation? And the central objective of this article is to analyze the construction of the field of scientific knowledge that has innovation as an attribute of mechatronics, in addition to identifying the main elements of mechatronics from quantitative bibliometric analysis and scientific collaboration networks and analyzing the publications of the sample from an analysis framework”. Even more, authors need to explicitly say what is the analysis framework.
RESPONSE: Thank you for your clarification regarding what was not being clear. As you can notice at the Introduction section, we have inserted a paragraph where we enlightenment and stress the contribution motive of this study as per your pertinent request. Being the innovation a very relevant attribute present in mechatronic constitution as a science, researching your association enable us to understand and get to know how a science develops from attributes such as innovation and its lateral areas. The modification is found on the page 3:
“Being the innovation a very relevant attribute present in mechatronic constitution as a science, researching your association enable us to understand and get to know how a science develops from attributes such as innovation and its lateral areas.And the central objective of this article is to analyze the constitution of the field of mechatronics based on the scientific knowledge that has innovation as an attribute of mechatronics. Our initial point was a framework for mechatronics and innovation from classical literature. Utilizing quantitative bibliometric analysis, it was possible to identify scientific collaboration networks, and by analyzing the sample publications with the initial framework, we mapped the main elements of mechatronics and the innovation perspectives that it reinforces in the scientific literature for the two largest available databases, Web of Science and Scopus”.
- The explanation about David Allam Brandley is very confusing and hard to follow. Authors state “Note that David Allan Bradley has two distinct forms of name entry: Bradley D. and Bradley D.A. As extracted from Scopus and Web of Science, your two author entries are valid and used to designate you in publications, so there are two clusters associated with the respective entries.” Who is your two authors and designate you? The next paragraph is also very confusing. Are the authors taking about two different authors or not?
RESPONSE: About David Allan Bradley, yes, both entries were considered for the same author, as per literature predictions on bibliometrics and information science. The motives for maintaining the two entries: David A. Bradley and David Bradley, have been rewritten as per your request. The excerpt can be found on page 48:
“To ensure that we are dealing with the scientific production of the same author, it is necessary to consider all possible entries in primary and secondary sources. David Allan Bradley, a single author, has two distinct ways of entering his name in the analyzed sample: Bradley D. and Bradley D.A. As extracted from Scopus and Web of Science, these author entries are valid and used to designate you in publications, so there are two clus-ters Bradley D. and Bradley D.A. associated with a single entry for author David Allan Bradley. Whose research underlies the variables of the qualitative analysis framework.
Although it is possible to change the author's name to a single format in the VOS Viewer input file, we chose to keep both formats because: i) it shows that the same author can be counted differently, which consists of at least point of attention in bibliometrics to understand when this happens; ii) if we change the file submitted to VOS Viewer in the author field due to the reduction in the number of characters, it reduces the cluster con-figuration, by deleting the way the author was cited in other works. Therefore, it was decided to maintain the original configuration, where the two Bradley clusters are arranged to designate their publications.”
- The references are still not cited properly.It is not about the format, it is that the articles and proceedings that were part of the analysis are not considered as part of references, they must be considered as supporting material or annex.
RESPONSE: Thank you for clarifying. We've reached out the the magazine editor and we were oriented to insert the 646 publications as attachments in a renumbered and alphabetical order, so we did.
- English also needs to be revised. There are parts that are confusing and hard to follow.
RESPONSE: We hire a language proofreader and due to the length and complexity of the material it took longerthan we're expecting .
We hope to have comply with all your request accordingly. Beforehand we appreciate all you effort and attention that has been granted to us.
